# Solving Satisfiability Modulo Counting Exactly with Probabilistic Circuits

**Jinzhao Li** [* 1]  **Nan Jiang** [* 1]  **Yexiang Xue** [1]

## Abstract

Satisfiability Modulo Counting (SMC) is a recently proposed general language to reason about problems integrating statistical and symbolic Artificial Intelligence. An SMC problem is an extended SAT problem in which the truth values of a few Boolean variables are determined by probabilistic inference. Approximate solvers may return solutions that violate constraints. Directly integrating available SAT solvers and probabilistic inference solvers gives exact solutions but results in slow performance because of many back-and-forth invocations of both solvers. We propose KOCO-SMC, an integrated exact SMC solver that efficiently tracks lower and upper bounds in the probabilistic inference process. It enhances computational efficiency by enabling early estimation of probabilistic inference using only partial variable assignments, whereas existing methods require full variable assignments. In the experiment, we compare KOCO-SMC with currently available approximate and exact SMC solvers on large-scale datasets and real-world applications. The proposed KOCO-SMC finds exact solutions with much less time.

## 1. Introduction

Symbolic and statistical Artificial Intelligence (AI) are two foundations with distinct strengths and limitations. Symbolic AI, exemplified by SATisfiability (SAT) and constraint programming, excels in constraint satisfaction but cannot handle probability distributions. Statistical AI captures probabilistic uncertainty but does not guarantee satisfying symbolic constraints. Integrating symbolic and statistical AI remains an open field and has gained much research attention (d'Avila Garcez et al., 2015; Li et al., 2023; Dennis et al., 2023; Sheth & Roy, 2024).

---
[*]Equal contribution [1]Department of Computer Science, Purdue University, IN, USA. Correspondence to: Jinzhao Li <li4255@purdue.edu>.

*Proceedings of the 41st International Conference on Machine Learning*, Vancouver, Canada. PMLR 267, 2025. Copyright 2025 by the author(s).

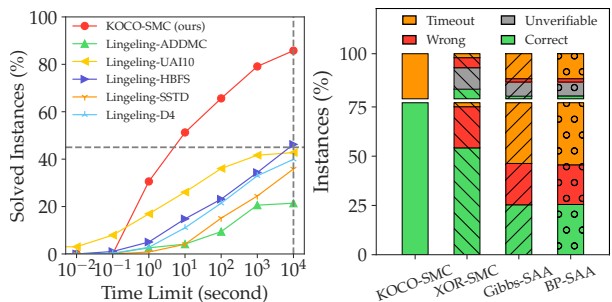

*Figure 1.* **(Left)** Compared to exact solvers, our KOCO-SMC solves 45% of SMC problems within a 10-second time limit, whereas baselines require up to 3 hours for a similar rate of completion. **(Right)** Compared with approximate methods, KOCO-SMC finds exact solutions for most cases (lower segment), whereas the approximate solvers may handle some hard instances (upper segment) but often yield lower-quality results.

Satisfiability Modulo Counting (SMC) is a recently proposed general language to reason about problems integrating statistical and symbolic AI (Fredrikson & Jha, 2014; Li et al., 2024). An SMC problem is an extended SAT problem in which the truth values of certain Boolean variables are determined through probabilistic inference, which assesses whether the marginal probability meets the given requirements. Solving SMC problems poses great challenges since they are NP[PP]-complete (Park & Darwiche, 2004).

Taking robust supply chain design as an example (Figure 2), a manager must choose a route on the road map to ensure sufficient materials for production, while accounting for stochastic events such as natural disasters. This problem necessitates both symbolic reasoning to find a satisfiable route and statistical inference to ensure the selected roads are robust to stochastic natural disasters. The SMC formulation is further detailed in Section 3.1. Slightly modified problems can be found in many real-world applications, including vehicle routing (Toth & Vigo, 2002), internet resilience (Israeli & Wood, 2002), social influence maximization (Kempe et al., 2005), energy security (Almeida et al., 2019), etc.

Several approximate SMC solvers have been proposed (Kleywegt et al., 2002; Li et al., 2024). Among them, Sample Average Approximation (SAA)-based methods (Kleywegt et al., 2002) are the most widely adopted,

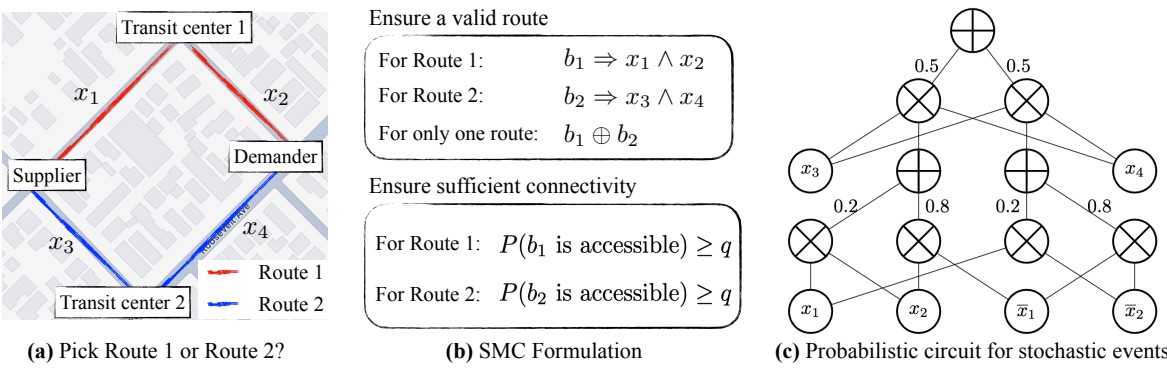

**(a)** Pick Route 1 or Route 2?    **(b)** SMC Formulation    **(c)** Probabilistic circuit for stochastic events

*Figure 2.* Formulation of the robust supply chain problem into an SMC problem. **(a)** A road map containing 4 locations and the road between them. The connectivity of each road is denoted by a random variable $x_i$, where $x_i = \texttt{True}$ indicates the corresponding road is selected. **(b)** Model the supply routine planning as an SMC problem. **(c)** The probability of every connectivity situation is represented as the Probabilistic Circuit. Each $x_i$ or $\overline{x}_i$ node denotes a leaf node that encodes a Bernoulli distribution. The symbols $\oplus$ and $\otimes$ represent the sum and product nodes, respectively. The values next to the edges are weights for the sum nodes.

estimating marginal probabilities using sample means. Another approach, XOR-SMC (Li et al., 2024) provides a constant-factor approximation guarantee by leveraging XOR-sampling to estimate marginals. Nevertheless, even the solutions found by XOR-SMC may violate a fraction of constraints because of the approximate bound. Overall, approximate solvers may be insufficient for certain scenarios where complete constraint satisfaction is essential.

In the absence of existing exact SMC solvers, an intuitive method to solve SMC problems exactly requires integrating SAT solvers and probabilistic inference solvers. Specifically, the SAT solver first gives a feasible variable assignment for the Satisfiability part, which is then evaluated by a probabilistic inference solver. This setup leads to excessive back-and-forth communication between the two solvers. A motivating example is in Section 3.1. For unsatisfiable instances, in particular, such exact solvers may exhaustively enumerate all possible assignments before concluding unsatisfiability, resulting in significant computational overhead.

We introduce KOCO-SMC, an exact and efficient SMC solver, mitigating the extreme slowness typically encountered in unsatisfiable SMC problems. KOCO-SMC saves time by detecting the conflict early with partial variable assignments. The proposed Upper Lower Watch (ULW) algorithm tracks the upper and lower bounds of probabilistic inference when new variables are assigned. When these bounds violate the satisfaction condition—for example, if the upper bound of the probability falls below the required threshold—the conflict is recorded as a learned clause. This clause is then used to prune the search space, preventing redundant exploration in subsequent iterations.

In our experiments, we evaluate all existing approximate and exact solvers by creating a large-scale dataset containing 1,350 SMC problems–based on the UAI Competition

benchmark held between 2010 and 2022. Figure 1 shows the comparison with state-of-the-art solvers. Compared with exact solvers, KOCO-SMC scales the best. Our KOCO-SMC solves 45% of instances within 10 seconds, whereas baseline methods require 3 hours. Given a 3-hour runtime, our approach can solve 85% of instances. Compared with approximate solvers under a 10-minute time limit, KOCO-SMC reliably delivers exact solutions for most cases, whereas approximate solvers solve some problems (including a few hard ones that KOCO-SMC cannot) but may return results that do not fully satisfy constraints or cannot be verified.

To summarize, our main contributions are:

1. We propose KOCO-SMC, an efficient exact solver for SMC problems, integrating probabilistic circuits with ULW algorithm for effective conflict detection.

2. Experiments on large-scale datasets illustrate KOCO-SMC's superior performance compared to state-of-the-art approximate and exact baselines in solution quality or time efficiency.

3. In the case study, we also demonstrate the process of formulating real-world problems into SMC problems, and highlight the strong capability of our solver in addressing these problems[1].

## 2. Preliminaries

**Satisfiability Modulo Counting (SMC)** provides a general language to reason about problems integrating symbolic and statistical constraints (Fredrikson & Jha, 2014; Li et al., 2024). Specifically, the symbolic constraint is character-

---

[1]Code implementation is available at: `https://github.com/jil016/koco-smc`

ized by a Boolean formula $\phi$, and the statistical constraint is captured by constraints involving marginal probability $\sum f_j$.

Let lowercase letters be random variables (i.e., $x$, $y$, $z$, and $b$) and let bold symbols (i.e., $\mathbf{x}$, $\mathbf{y}$, $\mathbf{z}$ and $\mathbf{b}$) denote vectors of Boolean variables, e.g., $\mathbf{x} = (x_1, \ldots, x_N)$. All variables take binary values in $\{\texttt{False}, \texttt{True}\}$.

Given a formula $\phi$ for Boolean constraints and two sets of weighted functions, $\{f_j\}_{j=1}^{M}$ and $\{g_k\}_{k=1}^{K}$, representing discrete probability distributions, the SMC problem is to determine if the following formula is satisfiable over Boolean variables $\mathbf{x}$ and $\mathbf{b}$:

$$\phi(\mathbf{x}, \mathbf{b}), \text{where } b_j \Leftrightarrow \sum_{\mathbf{y}_j} f_j(\mathbf{x}, \mathbf{y}_j) \geq q_j \tag{1}$$
$$\text{or } b_j \Leftrightarrow \sum_{\mathbf{y}_j} f_j(\mathbf{x}, \mathbf{y}_j) \geq \sum_{\mathbf{z}_k} g_k(\mathbf{x}, \mathbf{z}_k)$$
$$\text{for } j = 1, \ldots, M$$

Each function $f_j$ (or $g_k$) is an unnormalized discrete probability function over Boolean variables $\mathbf{x}$ and $\mathbf{y}_j$ (respectively, $\mathbf{x}$ and $\mathbf{z}_k$). The summation $\sum f_j$ and $\sum g_k$ compute the marginal probabilities, where $\mathbf{y}_j$ and $\mathbf{z}_k$ are latent variables and will be marginalized out. Thus, only $\mathbf{x}$ and $\mathbf{b}$ are decision variables.

Each $b_j$ is referred to as a *Probabilistic Predicate*, which is evaluated as true if and only if the inequality over the marginalized probability is satisfied. Each probabilistic constraint is in the form of either (1) the marginal probability surpassing a given threshold $q_j$, or (2) one marginal probability being greater than another. Note that the biconditional "$\Leftrightarrow$" can be relaxed to "$\Rightarrow$" or "$\Leftarrow$", and the "$\geq$" inequality can be generalized to "$=, >$", or to the reversed direction. In this paper, we focus on the form given in the first line of equation (1).

**Probabilistic Circuits (PCs)** are a broad class of probabilistic models that support a wide range of exact and efficient inference tasks (Darwiche, 2002; 1999; Poon & Domingos, 2011; Rahman et al., 2014; Kisa et al., 2014; Dechter & Mateescu, 2007; Vergari et al., 2020; Peharz et al., 2020). Formally, a PC is a computational graph that encodes a probability distribution $P(\mathbf{x})$ over a set of random variables $\mathbf{x}$. The graph consists of three types of nodes: leaf nodes, product nodes, and sum nodes. Each node represents a distribution over a subset of the variables.

Figure 2(c) illustrates an example PC over four variables. A leaf node $u$ encodes a tractable univariate distribution $P_u(x_i)$ over a single variable $x_i$, such as a Gaussian or Bernoulli distribution. A product node $u$ defines a factorized distribution $P_u(\mathbf{x}) = \prod_{v \in \texttt{ch}(u)} P_v(\mathbf{x})$, where $\texttt{ch}(u)$ denotes the children of node $u$. A sum node $u$ represents a mixture distribution $P_u(\mathbf{x}) = \sum_{v \in \texttt{ch}(u)} w_v P_v(\mathbf{x})$, where $w_v$ are normalization weights associated with each child $v$.

The root node of PCs encodes the full joint distribution over all variables.

When equipped with certain structural properties, PC enables efficient inference tasks—including computing partition functions, marginal probabilities, and MAP estimates—that scale polynomially with the graph size (Darwiche & Marquis, 2002; Choi et al., 2022).

## 3. Methodology

### 3.1. Motivation

We use robust supply chain design as a motivating example to illustrate the limitations of intuitive exact SMC solvers, which we construct as baselines in the absence of existing exact solvers. In Figure 2(a), the task is to deliver sufficient materials from suppliers to demanders on a road map. Various random events, such as natural disasters and car accidents, may affect road connectivity. The goal is to pick a route with a sufficient road connection probability.

Let $x_i$ denote the $i$-th road segment on the map, for $i = 1, \ldots, 4$. In the Boolean formula $\phi$, the assignment $x_i = \texttt{True}$ indicates that road segment $x_i$ is selected. The uncertainty due to random events is modeled by a joint probability distribution over all road segments, $P(x_1, x_2, x_3, x_4)$. Figure 2(c) uses a probabilistic circuit to represent $P(x_1, x_2, x_3, x_4)$. We represent the route selection using Boolean variables $b_1$ and $b_2$, where $b_1 = \texttt{True}$ corresponds to choosing route 1, which requires $x_1 = x_2 = \texttt{True}$. The probability that route 1 is fully connected—that is, all required segments are accessible—is given by the marginal probability:

$$P(b_1 \text{ is assessible}) = \sum_{x_3, x_4} P(x_1 = x_2 = \texttt{True}, x_3, x_4).$$

Let $q \in [0, 1]$ denote the minimum required probability for reliable connectivity along the selected route. The goal is to select either route 1 ($b_1 = \texttt{True}$) or route 2 ($b_2 = \texttt{True}$) such that the corresponding route's connectivity probability exceeds $q$. This can be summarized as the SMC problem:

$$\phi(\mathbf{x}, \mathbf{b}) = \underbrace{(b_1 \oplus b_2)}_{(a)} \wedge \underbrace{(b_1 \Rightarrow x_1 \wedge x_2)}_{(b)} \wedge \underbrace{(b_2 \Rightarrow x_3 \wedge x_4)}_{(c)},$$

$$\text{where} \quad \underbrace{b_1 \Leftrightarrow \sum_{x_3, x_4} P(x_1, x_2, x_3, x_4) \geq q,}_{(d)}$$

$$\underbrace{b_2 \Leftrightarrow \sum_{x_1, x_2} P(x_1, x_2, x_3, x_4) \geq q,}_{(e)}$$

where $\oplus$ is the logical "exclusive or" operator. In part (a), the constraint ensures that only one route is selected. In part

(b), the constraint indicates that: if route 1 is selected, both $x_1$ and $x_2$ must be assigned True. Part (c) applies a similar condition for route 2. In part (d), $\sum_{x_3, x_4} P(x_1, x_2, x_3, x_4)$ marginalizes out $x_3$ and $x_4$, representing the probability of route 1's connectivity condition under random natural disasters. Part (e) is analogous to part (d).

Since no general exact SMC solver currently exists, solving SMC problems exactly requires combining tools from different domains. Assume $q = 0.5$, we have:

1. It first uses an SAT solver to solve the Boolean SAT problem $\phi(\mathbf{x}, \mathbf{b})$ and proposes a solution, e.g., $x_1 = x_2 = b_1 = $ True, $x_3 = x_4 = b_2 = $ False. It implies that only route 1 is selected.

2. Then, it infers the marginal probability $\sum_{x_3, x_4} P(x_1 = x_2 = $ True$, x_3, x_4) = 0.1 < q$, which violates the probabilistic constraint. The detailed calculation is shown in Figure 7.

3. Since variables within the probabilistic constraint cause a conflict, it adds the negated clause $\neg(x_1 \land x_2 \land b_1)$ to formula $\phi$ to omit this assignment in the future, and then returns to step 1 to find a new assignment.

The process exhibits a sequential dependency between the SAT solver and probabilistic inference, wherein each component must await the completion of the other. This mutual blocking results in suboptimal computational efficiency, and in the worst-case scenario, the SAT solver is required to exhaustively enumerate all feasible solutions.

To address this issue, our KOCO-SMC immediately detects a conflict upon the partial assignment $x_1 = $ True, saving time by avoiding further assignments to the remaining variables. Although $x_2$ remains unassigned, the maximum achievable probability under any completion of the assignment is already below the threshold $q$:

$$\max_{x_2} \sum_{x_3, x_4} P(x_1 = \text{True}, x_2, x_3, x_4) = 0.1 < q.$$

This should trigger an immediate conflict, rather than deferring conflict detection until the SAT solver assigns $x_2$. As a result, KOCO-SMC achieves greater efficiency than existing exact SMC solvers by pruning infeasible branches earlier in the search process.

### 3.2. Main Pipeline of KOCO-SMC

This section outlines the detailed procedure of the proposed KOCO-SMC for solving SMC problems. As SMC problems extend standard SAT by incorporating probabilistic constraints, KOCO-SMC builds upon and extends the classical Conflict-Driven Clause Learning framework (Silva & Sakallah, 1996; Eén & Sörensson, 2003), which comprises

four components: variable assignment, propagation, conflict clause learning, and backtracking. Each component is systematically adapted to handle probabilistic constraints. A high-level overview is provided in Algorithm 1.

**Compilation.** Initially, knowledge compilation transforms all probability distributions into PCs with smooth and decomposable properties. This can be achieved by advanced tools (Darwiche & Marquis, 2002; Darwiche, 2004; Lagniez & Marquis, 2017). An example of compiled PCs is provided in Figure 2(c).

**Variable Assignment.** Pick one variable among the remaining free variables and assign it with a value in {True, False}. Practical heuristics on the choice of variable and value to accelerate the whole process can be found in Moskewicz et al. (2001); Eén & Sörensson (2003); Hamadi et al. (2009).

**Propagation.** Given partial variable assignments, this step simplifies the formula by propagating their logical implications. For Boolean constraints, *unit propagation* (Zhang & Stickely, 1996) is used to infer additional variable assignments and detect conflicts. For example, consider the Boolean formula $\phi = (x_1 \lor \neg x_2) \land (x_2 \lor x_3)$. If we assign $x_1 = $ False, unit propagation forces $x_2 = $ False to satisfy the clause $(x_1 \lor \neg x_2)$, which subsequently propagates $x_3 = $ True to satisfy the clause $(x_2 \lor x_3)$. This cascading effect significantly improves the efficiency of solving Boolean constraints.

However, existing propagation techniques are specifically designed for purely Boolean formulas. Extending this process to incorporate probabilistic constraints—by extracting useful information from partial assignments and efficiently detecting conflicts—remains a challenging open problem. To address this, we propose the Upper-Lower Watch (ULW) method (see Section 3.3), a novel propagation technique for probabilistic constraints. ULW leverages tractable probabilistic circuits, enabled by advances in modern knowledge compilers, to efficiently track the upper and lower bounds of marginal probabilities.

**Conflicts Clause Learning.** A *conflict* occurs when the current partial assignment violates either a Boolean constraint or a probabilistic constraint, indicating that the current branch of the search cannot lead to a satisfying solution. In such cases, a *learned clause* is derived and added to the Boolean formula to prevent the solver from revisiting the same conflicting assignment in the future. This mechanism enables KOCO-SMC to effectively prune the search space and significantly accelerate the solving process.

When a conflict is detected within a Boolean clause, there are existing techniques to add a learned clause to the original Boolean formula, preventing the same conflict from occurring in the future (Hamadi et al., 2009).

When a conflict arises from a probabilistic constraint, KOCO-SMC generates a learned Boolean clause that captures the root cause of the violation. This clause is constructed by negating the current partial assignment responsible for making the constraint unsatisfiable. For example, consider the probabilistic constraint $\sum_{x_3,x_4} P(x_1 = \texttt{True}, x_2, x_3, x_4) < q$, which is unsatisfiable under the current assignment $x_1 = \texttt{True}$. KOCO-SMC derives the learned clause $\neg x_1$. This clause is then added to the Boolean formula, resulting in an updated constraint $\phi \wedge (\neg x_1)$, which prevents the solver from repeating the same conflict and eliminates the need to re-evaluate the same probabilistic inference in future branches.

**Backtracking.** This step undoes variable assignments when a conflict is detected, enabling the solver to backtrack and explore alternative branches of the search space.

### 3.3. Upper-Lower Watch for Conflict Detection in Probabilistic Constraints

The satisfaction or conflict of a probabilistic constraint is determined by the involved marginal probability. By maintaining an interval that bounds this marginal probability and refining it with each new variable assignment, we can detect satisfiability or conflict early when the range significantly deviates from the threshold.

Let $\mathbf{x}_{\texttt{assigned}}$ be the assigned variables, $\mathbf{x}_{\texttt{rem}}$ denote the unassigned variables, and $\mathbf{y}$ be the marginalized-out latent variables. Determining the range of a marginal probability involves estimating the appropriate interval $[\texttt{LB}, \texttt{UB}]$, such that for all possible values assigned to $\mathbf{x}_{\texttt{rem}}$:

$$\texttt{LB} \leq \sum_{\mathbf{y}} P(\mathbf{x}_{\texttt{assigned}}, \mathbf{x}_{\texttt{rem}}, \mathbf{y}) \leq \texttt{UB}, \qquad (2)$$

where $\texttt{UB}$ (resp. $\texttt{LB}$) is an estimated "upper bound" (resp. "lower bound") of the probabilistic constraint given current partial assignment $\mathbf{x}_{\texttt{assigned}}$.

The proposed Upper-Lower Watch (ULW) algorithm monitors both values to track constraint violations. We show that computing sufficiently tight bounds can be reduced to a traversal of these circuits.

Specifically, each node $v$ in the probabilistic circuit represents a distribution $P_v$ over variables covered by its children. Our ULW algorithm associates each node with an upper bound $\texttt{UB}(v)$ and a lower bound $\texttt{LB}(v)$ for the marginal probability of $P_v$ under the current assignment $\mathbf{x}_{\texttt{assigned}}$. Therefore, the $\texttt{UB}(r)$ and $\texttt{LB}(r)$ at the root node $r$ are the upper and lower bounds of the whole probability.

To initialize or update the bounds, we traverse the probabilistic circuits in a bottom-up manner. The update rule for the leaf nodes is:

• For a leaf node $v$ over an assigned variable $x \in \mathbf{x}_{\texttt{assigned}}$,

where variable $x$ is assigned to $\texttt{val}$, update $\texttt{UB}(v) = \texttt{LB}(v) = P_v(x = \texttt{val})$.

• For a leaf node $v$ over an remaining variable $x \in \mathbf{x}_{\texttt{rem}}$, update $\texttt{UB}(v) = \max\{P_v(x = \texttt{True}), P_v(x = \texttt{False})\}$ and $\texttt{LB}(v) = \min\{P_v(x = \texttt{True}), P_v(x = \texttt{False})\}$.

• For a leaf node $v$ over $y \in \mathbf{y}$ (the variable to be marginalized), update $\texttt{UB}(v) = \texttt{LB}(v) = 1$.

Let $\texttt{ch}(v)$ be the set of child nodes of $v$. Intermediate nodes, i.e., product nodes and sum nodes, can be updated by:

• For a product node $p$, update $\texttt{UB}(p) = \prod_{u \in \texttt{ch}(p)} \texttt{UB}(u)$, and $\texttt{LB}(p) = \prod_{u \in \texttt{ch}(p)} \texttt{LB}(u)$.

• For a sum node $s$, $\texttt{UB}(s) = \sum_{u \in \texttt{ch}(s)} w_u \texttt{UB}(u)$, and $\texttt{LB}(s) = \sum_{u \in \texttt{ch}(s)} w_u \texttt{LB}(u)$. Here, $w_u$ is the weight associated with each child node $u$.

During initialization, the entire probabilistic circuit is traversed once to compute the bounds for every node. As the solving process proceeds, assigning a free variable triggers updates only along the path from the affected leaf nodes to the root, ensuring computational efficiency. The correctness of the whole execution procedure is guaranteed by the smoothness and decomposability properties of probabilistic circuits. A former justification is provided in Lemma 3.1.

The bounds at the root node $r$ correspond to the bounds of the marginal probability in the probabilistic constraint. These estimated bounds are then used for conflict detection. Basically, if the lower bound $\texttt{LB}(r)$ at the root exceeds the threshold $q$ in Equation (1), the constraint is guaranteed to be satisfied. Conversely, if the upper bound $\texttt{UB}(r)$ is less than $q$, the constraint is unsatisfiable.

**Lemma 3.1.** *Let probabilistic circuit $P(\mathbf{x}, \mathbf{y})$ defined over Boolean variables $\mathbf{x}$ and $\mathbf{y}$. If the probabilistic circuit satisfies the smooth and decomposable property, our ULW guarantees that Equation 2 holds. The equality is attained for both $\texttt{LB}$ and $\texttt{UB}$ when all variables are assigned.*

*Sketch of Proof.* The result follows from applying the properties of smooth and decomposable probabilistic circuits to the marginal probability inference problem. For a detailed proof, please refer to Appendix B. $\square$

## 4. Related Works

**Satisfiability Problems.** Satisfiability (SAT) determines whether there exists an assignment of truth values to Boolean variables that makes the entire logical formula true. Numerous SAT solvers show great performance in various applications (Moskewicz et al., 2001; Silva & Sakallah, 1999; Eén & Sörensson, 2003; Hamadi et al., 2009).

---

**Algorithm 1** Solving Satisfiability Modulo Counting Exactly with Probabilistic Circuits.

---

**Input:** Boolean formula $\phi$; $M$ weighted functions $\{f_j\}_{j=1}^M$ and thresholds $\{q_j\}_{j=1}^M$; Boolean variables $\mathbf{x} = (x_1 \ldots, x_N)$ and $\mathbf{b} = (b_1, \ldots, b_M)$.

**Output:** Satisfiability, variable assignment.

1: Compile $M$ weighted functions $\{f_j\}_{j=1}^M$ into probabilistic circuits $\{C_j\}_{j=1}^M$.       ▷ Compilation
2: **loop**
3:     Assign a free variable $x$ to a value in $\{\texttt{True}, \texttt{False}\}$;       ▷ Variable assignment
4:     Perform unit propagation on $\phi$.
5:     **for** each probabilistic constraint $C_j$ **do**
6:         Update bounds for probabilistic constraint $C_j$.       ▷ ULW algorithm
7:         Detect conflicts by comparing bounds with threshold $q_j$.
8:     **if** no conflict is detected **then**
9:         **if** all variables are assigned **then**
10:             **return** SAT, variable assignments.
11:     **else**
12:         Propose a learned clause $c_l$ and update Boolean formula $\phi \leftarrow \phi \wedge c_l$.       ▷ Clause learning
13:         **if** no variable has been assigned **then**
14:             **return** UNSAT, no assignment.
15:         **else**
16:             Undo assignments.       ▷ Backtracking

---

Conflict-Driven Clause Learning (CDCL) (Silva & Sakallah, 1996) is a modern SAT-solving framework that has been widely applied. The process begins by making decisions to assign values to variables and propagating the consequences of these assignments. If a conflict is encountered, i.e., a clause is unsatisfied, the solver performs conflict analysis to learn a new clause that prevents the same conflict in the future. The solver then backtracks to an earlier decision point, and the process continues. Through clause learning and backtracking, CDCL improves efficiency and increases the chances of finding a solution or proving unsatisfiability. Our KOCO-SMC extends every component in the CDCL framework to handle probabilistic constraints.

**Probabilistic Inference and Model Counting.** Probabilistic inference encompasses various tasks, such as calculating conditional probability, marginal probability, maximum a posteriori probability (MAP), and marginal MAP (Cheng et al., 2012). Each of them is essential in fields like machine learning, data analysis, and decision-making processes. Model counting calculates the number of satisfying assignments for a given logical formula, and is closely related to probabilistic inference (Gomes et al., 2006; Achlioptas & Theodoropoulos, 2017). In discrete probabilistic models, computing probabilities can be translated to model counting (Chavira & Darwiche, 2008).

Our KOCO-SMC efficiently tracks upper and lower bounds for probabilistic constraints with partial variable assignments. In literature, Dubray et al. (2024); Ge & Biere (2024) compute approximate bounds for probabilistic constraints based on the DPLL algorithm. Marinescu et al. (2014); Ping et al. (2015); Choi et al. (2022) provide exact bounds by solving marginal MAP problems, yet they are more time-consuming than our method.

**Probabilistic Circuit** with specific structural properties, i.e., decomposability (Darwiche, 2001a; 1999), smoothness (Darwiche, 2001b), and determinism (Darwiche & Marquis, 2002), enable efficient probability inferences, scaling polynomially with circuit size. For example, partition functions and marginal probabilities are computed efficiently due to decomposability and smoothness, MAP requires determinism for maximization, and Marginal MAP further requires Q-determinism (Choi et al., 2022).

The process of transforming a probability distribution into a probabilistic circuit with a specific structure is referred to as *knowledge compilation* (Darwiche, 1999; 2001b;a; Darwiche & Marquis, 2002). Several knowledge compilers, such as ACE (Darwiche & Marquis, 2002), C2D (Darwiche, 2004), and D4 (Lagniez & Marquis, 2017), have been developed to convert discrete distributions into tractable PCs for various probabilistic inference tasks.

**Specialized Satisfiability Modulo Counting.** Stochastic Satisfiability (SSAT) (Papadimitriou, 1985) can encode SMC problems with a Boolean constraint and one single probabilistic constraint by integrating Boolean SAT with probabilistic quantifiers. Advances in SSAT solvers (Lee et al., 2017; 2018; Fan & Jiang, 2023; Cheng et al., 2024) have improved their efficiency, but these solvers remain limited to problems with a single probabilistic constraint.

Stochastic Constraint Optimization Problems can model

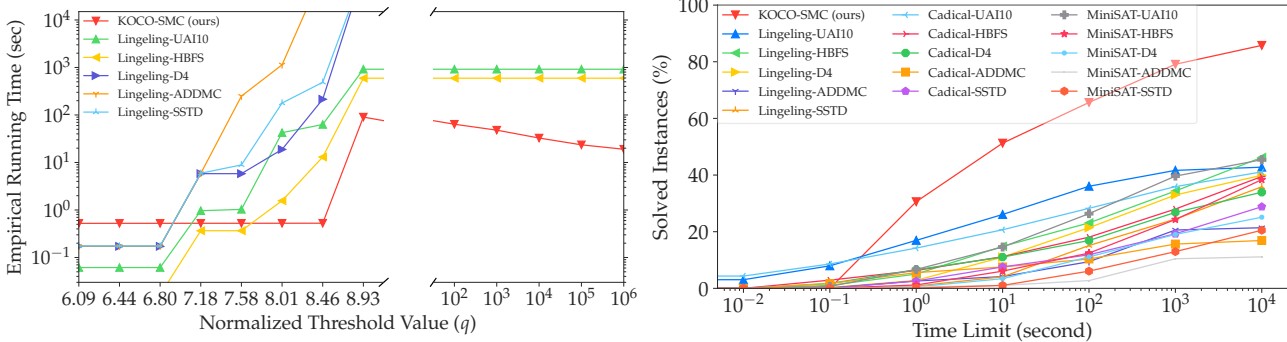

*Figure 3.* (**Left**) The running time (horizontal axis) of different solvers on a selected SMC instance with varying thresholds (vertical axis). Our method is slower before the critical point due to the overhead of compilation time, but becomes much faster than the baselines beyond it due to early conflict detection. Additional results are in appendix Figures 13-15. (**Right**) The percentage of instances from the entire dataset solved within the time limit. Compared to exact solvers, KOCO-SMC solves 80% of SMC problems in 20 minutes, whereas baselines solve at most 40% of instances in a 3-hour time limit.

SMC problems by incorporating stochastic constraints. Existing approaches solve them using techniques from Mixed-Integer Linear Programming or Constraint Programming (Latour et al., 2019; 2017).

## 5. Experiments

In the experiments, Figure 3 shows KOCO-SMC 's superior time efficiency compared to available exact solvers. Figure 4 demonstrates KOCO-SMC 's advantage in finding exact solutions over approximate solvers. Figure 5 shows that the proposed ULW algorithm accelerates solving SMC problems. Finally, Figure 6 demonstrates that KOCO-SMC can efficiently handle two real-world applications.

### 5.1. Experiment Settings

**Dataset.** We fix the number of probabilistic constraints to one. For the weighted function $f$, we use benchmark instances from the partition function task in the Uncertainty in Artificial Intelligence (UAI) Challenges, held between 2010 and 2022. We retain 50 instances defined over binary variables and group them into six categories: *Alchemy* (1 model), *CSP* (3 models), *DBN* (6 models), *Grids* (2 models), *Promedas* (32 models), and *Segmentation* (6 models). For the Boolean formula $\phi$, we generate 9 random 3-coloring map problems using CNFgen (Lauria et al., 2017). These instances involve binary encodings with variable counts ranging from 75 to 675. Unless noted otherwise, each task uses three thresholds $q$ obtained by multiplying the model's partition function by $10^{-20}$, $10^{-10}$, and $10^{-1}$. This yields 1,350 data points in total.

**Baselines** We consider several approximate SMC solvers and exact SMC solvers. For the approximate solver, we include the Sampling Average Approximation (SAA) (Kleywegt et al., 2002)-based method. Specifically, we use Lin-

geling SAT solver (Heule et al., 2019) to enumerate solutions and estimate $\sum_{\mathbf{y}} f(\mathbf{x}_f, \mathbf{y})$ using sample means, which enables approximate inference of marginal probabilities. We Gibbs Sampler (Shapiro, 2003) (Gibbs-SAA) and Belief Propagation (BP-SAA) (Ding & Xue, 2020) to draw samples. We also include XOR-SMC (Li et al., 2024), an approximated solver specifically for SMC problems.

The exact solver baseline is composed of an exact SAT solver and probabilistic inference solvers. This approach first identifies a solution to the Boolean formula and then verifies it with the probabilistic constraints. For the Boolean SAT solver, we use Lingeling (Heule et al., 2019), CaDiCal (Biere et al., 2024), MiniSAT (Eén & Sörensson, 2003) for their superior performance. For probabilistic inference, we use the UAI2010 winning solver implemented in libDAI (Mooij, 2010) (UAI10) and the solver based on the hybrid best-first branch-and-bound algorithm (HBFS) developed by Toulbar2 (Cooper et al., 2010). Due to the underlying connection between probabilistic inference and weighted model counting, we also include model counters from recent Model Counting competitions (Fichte et al., 2021) from 2020 to 2023: D4 solver (Lagniez & Marquis, 2017), ADDMC (Dudek et al., 2020), and SSTD (Korhonen & Järvisalo, 2023).

**Implementation of KOCO-SMC.** We applied ACE (Darwiche & Marquis, 2002) as the knowledge compiler. The CDCL skeleton of KOCO-SMC is implemented on top of MiniSAT (Eén & Sörensson, 2003), for its easily extensible structure. Appendix D collects the detailed experiment settings.

### 5.2. Result Analysis

**Comparison with Exact Solvers.** We begin by examining how exact SMC solvers perform on problems with varying

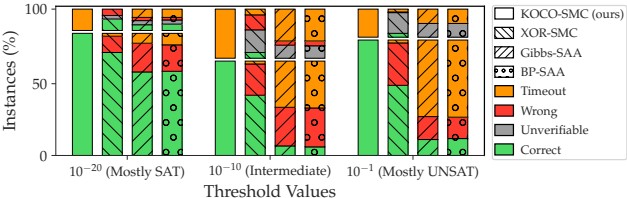

*Figure 4.* Comparison of KOCO-SMC and approximate solvers on datasets partitioned by threshold values. Each gap marks the point at which instances become exceedingly difficult for KOCO-SMC. Divided by this gap, we show the performance of approximate solvers accordingly. KOCO-SMC solves many instances exactly (lower segment), but it can time out on the complex cases (upper segment). Approximate solvers may produce answers to these challenging problems, but at the expense of solution quality.

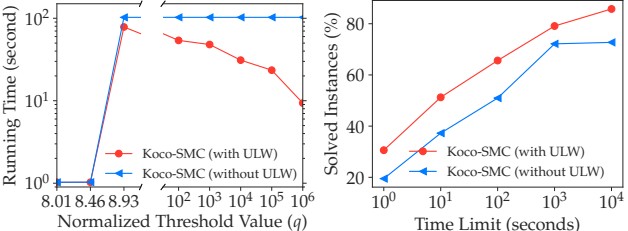

*Figure 5.* The ULW in KOCO-SMC is crucial for speeding up SMC problem solving. **(Left)** The running time with varying thresholds. ULW propagation accelerates KOCO-SMC by 10 times compared with KOCO-SMC without ULW when the threshold reaches $10^6$. **(Right)** The percentage of instances solved in 3 hours. ULW helps KOCO-SMC to solve more instances within a given running time.

numbers of satisfying solutions. This is done by adjusting the threshold value $q$ and measuring the corresponding solving time. As $q$ increases, satisfying assignments become increasingly rare, eventually resulting in unsatisfiable instances. Specifically, Figure 3(left) shows results for a single SMC instance under different $q$ values, where the Boolean constraint is derived from a 3-coloring problem on a $5 \times 5$ grid, and the probabilistic constraint is based on the `smokers-10.uai` instance from the UAI 2012 Competition.

At low threshold values, all solvers are able to quickly find satisfying assignments; however, KOCO-SMC incurs additional overhead due to the knowledge compilation step. As the threshold increases, satisfying assignments become rarer, leading to increased solving time across all methods.

Notably, when the threshold becomes sufficiently high such that the instance is unsatisfiable, KOCO-SMC exhibits reduced solving time, while the runtimes of other solvers remain high. This efficiency gain stems from KOCO-SMC's integrated ULW algorithm, which enables early detection of unsatisfiability—unlike baseline solvers, which must exhaustively enumerate all candidate assignments before concluding infeasibility.

The efficiency of KOCO-SMC is further validated by evaluating it on the full dataset. Figure 3 (right) shows the percentage of solved instances as a function of runtime. This figure extends the results in Figure 1 (right) by including additional configurations of KOCO-SMC using different Boolean SAT solvers—Lingeling (Lingeling-), MiniSAT (MiniSAT-), and CaDiCaL (CaDiCal-). Among all configurations and baseline methods, KOCO-SMC consistently achieves the best overall performance.

**Comparison with Approximate Solvers.** We compare KOCO-SMC with approximate solvers under a 10-minute time limit. For fairness, approximate solvers may run multiple times within this limit. Because approximate solvers cannot guarantee correctness, we classify their outputs as follows. If an output is verifiable, either because an exact solver supplies a reference solution or because the returned assignment can be checked by an exact model counter, we classify the output as correct or wrong; otherwise, it is unverifiable. Then for repeated runs, an instance is labeled *Correct* if any run produces a correct solution, *Timeout* if no run finishes in time, *Wrong* if all runs are wrong, and *Unverifiable* otherwise. The comparison between KOCO-SMC and approximate solvers is shown in Figure 4.

The SAA-based method provides rapid count estimates from samples. However, like the exact baselines, it can time out while searching for satisfying assignments, especially as the threshold rises and satisfying assignments become rarer. XOR-SMC efficiently reduces an SMC problem to a surrogate SAT instance with guarantees, but its solution quality degrades on the hardest (medium-threshold) cases, where many SMC instances lie near the SAT–UNSAT transition point. We found that XOR-SMC sometimes finds solutions that violate constraints. This is because it only checks the approximate bound. More careful parameter tuning and additional runs may improve its performance. Overall, KOCO-SMC solves most instances exactly but may time out on complex cases. Approximate solvers sometimes return answers for these difficult problems, yet their solutions are often of lower quality, being incorrect or cannot be verified.

**Effectiveness of Upper-Lower Bound Watch Algorithm.** In Figure 5(left), ULW propagation accelerates KOCO-SMC by 10 times compared with KOCO-SMC without ULW when the threshold reaches $10^6$. Figure 5(right) further demonstrates the contribution of ULW, where KOCO-SMC is 10 times faster than KOCO-SMC without ULW for SMC problems solvable in around 10 minutes.

### 5.3. Case Studies

**Application: Robust Supply Chain Design.** In a supply chain network, each supplier is represented as a node that

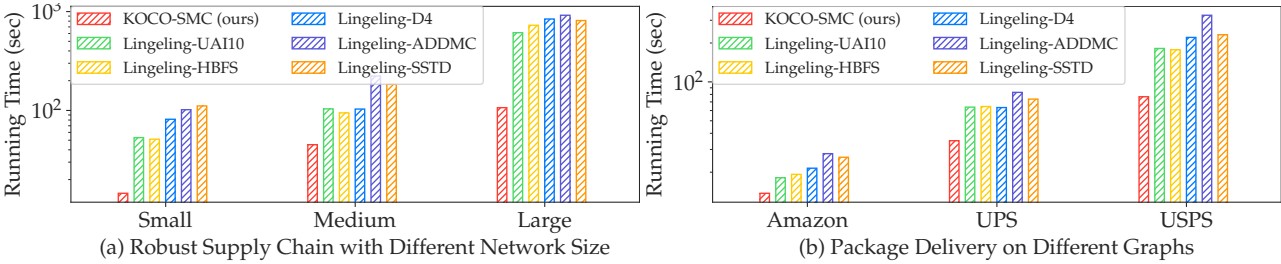

Figure 6. **(Left)** Running time of each method for identifying the best trading plan. All methods are tested on three real-world supply chain networks of different sizes. **(Right)** Running time for identifying the best delivery path. All methods are tested on three road maps of different sizes. Our KOCO-SMC finds all the exact satisfying solutions significantly faster.

purchases raw materials from upstream suppliers and sells products to downstream customers. The objective is to ensure a valid trading path from the source provider to the end demander, while maximizing the overall success probability in the presence of disruptions such as natural disasters. The problem formulation includes two types of constraints: (1) The Boolean constraint ensures the existence of a valid path from the source provider to the end demander. (2) The probabilistic constraint evaluates the reliability of roads, which may be disrupted by stochastic events such as natural disasters, traffic accidents, or political instability.

Let $x_i \in \{\texttt{True}, \texttt{False}\}$ represent the selection of trade between nodes connected by $i$-th edge, where $x_i = \texttt{True}$ if the trade is selected. Combining the requirements (1) and (2), we have the SMC formulation: $\phi(\mathbf{x}) \wedge \left( \sum_{\mathbf{y}} P(\mathbf{x}, \mathbf{y}) > q \right)$ where the marginal probability $\sum_{\mathbf{y}} P(\mathbf{x}, \mathbf{y})$ is the probability that all selected trades are carried out successfully and $q$ is the minimum requirement of successful probability.

We use 4-layer supply chain networks from the wheat-to-bread network with 44 nodes (Large) (Zokaee et al., 2017), where each layer represents a supplier tier. We also create synthetic networks with 20 (Small) and 28 (Medium) nodes. To identify the plan with the highest success probability, we incrementally raise the threshold $q$ from 0 to 1 by $1 \times 10^{-2}$ until the SMC problem becomes unsatisfiable. Detailed settings are in Appendix D.4. The running time for finding the best plan is shown in Figure 6(left). Our KOCO-SMC needs much less time than baselines to find the optimal plan.

**Application: Package Delivery.** The task is to find a path that visits all specified delivery locations exactly once while minimizing the probability of encountering heavy traffic (Hoong et al., 2012). Delivery locations and roads are modeled as nodes and edges in a graph, respectively. The problem involves two key components: (1) A Boolean constraint that ensures each location is visited exactly once, with road availability based on real-world data from Google Maps. (2) A probabilistic constraint that limits the likelihood of encountering heavy traffic on any road segment

to below a specified threshold. This probability accounts for factors such as congestion, extreme weather, and road construction.

Suppose there are $N$ delivery locations, and let $x_{i,j} = \texttt{True}$ denote that the $j$-th location is visited in the $i$-th position of the path. Combining constraints (1) and (2), we formulate the problem as an SMC instance: $\phi(\mathbf{x}) \wedge \left( \sum_{\mathbf{y}} P(\mathbf{x}, \mathbf{y}) < q \right)$, where $\mathbf{x} = \{x_{i,j} \mid i, j \in \{1, \ldots, N\}\}$ is the set of decision variables representing the path, and $\mathbf{y}$ is the set of latent environmental variables. The term $P(\mathbf{x}, \mathbf{y})$ denotes the probability of encountering heavy traffic given path $\mathbf{x}$ with environmental conditions $\mathbf{y}$. Detailed settings are provided in Appendix D.5.

We use three sets of delivery locations in Los Angeles: 8 Amazon Lockers, 10 UPS Stores, and 6 USPS Stores. The experimental graphs are: Amazon Lockers only (Amazon), Amazon Lockers plus UPS Stores (UPS), and the UPS graph extended with USPS Stores (USPS), with 8, 18, and 24 nodes, respectively. Traffic congestion probabilities are modeled using a Bayesian network trained on LA traffic data (West, 2020). Figure 6 (right) shows the runtime required to find the optimal delivery route. KOCO-SMC efficiently discovers high-quality delivery plans under real-world uncertainty.

## 6. Conclusion

We introduced KOCO-SMC for exact solving of Satisfiability Modulo Counting problems. Unlike existing methods that combine SAT solvers with model counters, KOCO-SMC employs an early conflict detection mechanism by comparing the upper and lower bounds of probabilistic inferences. Our Upper-Lower Watch algorithm efficiently tracks these bounds, enabling efficient solving. Experiments on large-scale datasets show that KOCO-SMC delivers higher solution quality than approximate solvers and significantly outperforms existing exact solvers in efficiency. Real-world applications further demonstrate its potential for solving practical problems.

## Acknowledgement

We thank all the reviewers for their constructive comments. This research was supported by NSF grant CCF-1918327, NSF Career Award IIS-2339844, DOE – Fusion Energy Science grant: DE-SC0024583.

## Impact Statement

Satisfiability Modulo Counting (SMC) extends traditional Boolean satisfiability by incorporating constraints that involve probability inference (model counting). This extension allows for solving complex problems where both logical and probabilistic constraints must be satisfied. SMC has wide applications in supply chain design, shelter allocation, scheduling problems, and many others in Operation Research. For example, in scheduling problems, SMC can ensure that selected schedules meet probabilistic events, while in shelter allocation, it can verify that the accessibility under random disasters is above a specified threshold.

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

# A. Probabilistic Inference in Probabilistic Circuits

Probabilistic inference in a probabilistic circuit can be highly efficient. Figure 7 illustrates a decomposable and smooth probabilistic circuit, where each node corresponds to a binary distribution. To compute $P(x_1 = x_3 = x_4 = \texttt{True}, x_2 = \texttt{False})$, set the values of nodes $x_1$, $\overline{x}_2$, $x_3$, and $x_4$ to 1, and set $x_2$ and $\overline{x}_1$ to 0. Finally, evaluate the root node to obtain the probability, which in this case is 0.1.

For the marginal probability, the circuit must be both decomposable and smooth to ensure correctness and efficiency. In Figure 7 (b), to calculate $P(x_3 = x_4 = \texttt{True})$, set the values of nodes $x_3$ and $x_4$ to 1, reflecting their assigned values. For the marginalized variables $x_1$ and $x_2$, set all associated nodes — $x_1$, $\overline{x}_1$, $x_2$, and $\overline{x}_2$ — to 1. Then evaluate the root node, which should yield a probability of 1.0.

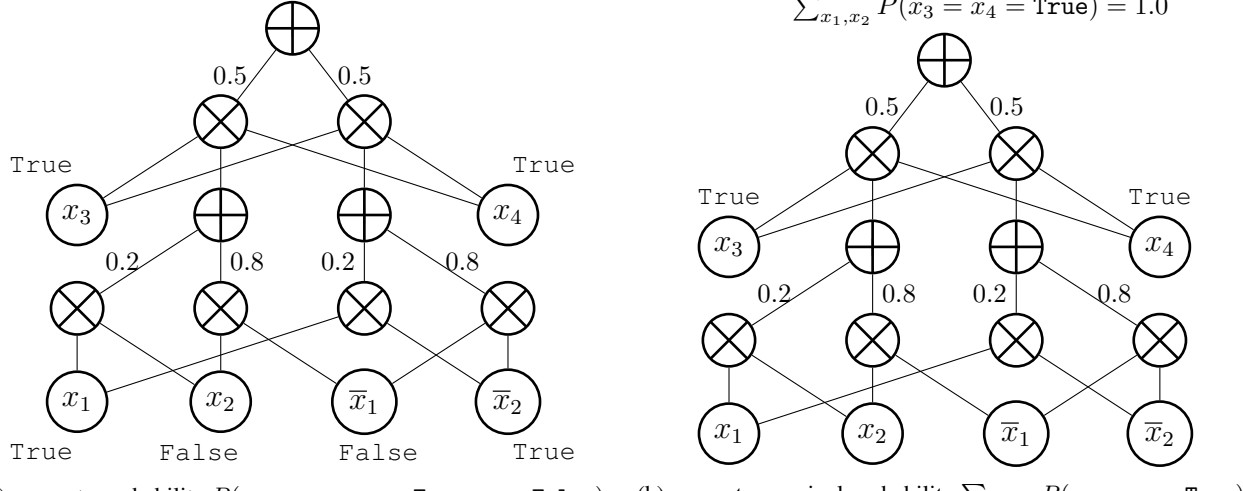

(a) compute probability $P(x_1 = x_3 = x_4 = \texttt{True}, x_2 = \texttt{False})$    (b) compute marginal probability $\sum_{x_1,x_2} P(x_3 = x_4 = \texttt{True})$.

*Figure 7.* Example of probability inference in a decomposable and smooth probabilistic circuit. To infer the probability $P(x_1 = x_3 = x_4 = \texttt{True}, x_2 = \texttt{False})$, set the value of nodes $x_1$, $\overline{x}_2$, $x_3$, and $x_4$ to 1. Also set the value of nodes $\overline{x}_1$ and $x_2$ to 0. The circuit evaluates to the probability 0.1. To infer the marginal probability $P(x_3 = x_4 = \texttt{True})$, set nodes $x_3$ and $x_4$ to 1. For the marginalized-out variables $x_1$ and $x_2$, set all related nodes to 1. The circuit evaluates to the marginal probability 1.0.

# B. Proof of Lemma 3.1

**Assumption B.1** (Smooth and Decomposable (Choi et al., 2022))**.** A *smooth* probabilistic circuit when all children of every sum node share identical sets of variables; A probabilistic circuit is *decomposable* if the children of every product node have disjoint sets of variables; Smoothness and decomposability enable *tractable* computation of any marginal probability query.

**Definition B.2.** Denote assigned variables in $\mathbf{x}$ as $\mathbf{x}_e$ and those not assigned as $\mathbf{x}_h$. The exact upper and lower bounds of the marginal probability with the partial variable assignment are $\max_{\mathbf{x}_h} \sum_{\mathbf{y}} P(\mathbf{x}_e, \mathbf{x}_h, \mathbf{y})$ and $\min_{\mathbf{x}_h} \sum_{\mathbf{y}} P(\mathbf{x}_e, \mathbf{x}_h, \mathbf{y})$.

*Proof.* To prove that $UB \geq \max_{\mathbf{x}_h} \sum_{\mathbf{y}} P(\mathbf{x}_e, \mathbf{x}_h, \mathbf{y})$ and $LB \leq \min_{\mathbf{x}_h} \sum_{\mathbf{y}} P(\mathbf{x}_e, \mathbf{x}_h, \mathbf{y})$, we need to relate the updating scheme with the marginal probability inference. We focus on the upper bound case ($UB$); the lower bound follows by a symmetric argument. Let $\mathbf{x}_h^* = \arg\max_{\mathbf{x}_h} \sum_{\mathbf{y}} P(\mathbf{x}_e, \mathbf{x}_h, \mathbf{y})$ denotes the assignment to $\mathbf{x}_h$ that maximizes the marginal probability. We now show that the updates in our ULW algorithm guarantee $UB \geq \sum_{\mathbf{y}} P(\mathbf{x}_e, \mathbf{x}_h^*, \mathbf{y})$ by recursion.

- **Base case.** Consider a leaf node $v$ corresponding to a single variable: (1) If $x \in \mathbf{x}_e$ is already assigned, then $UB(v) = P_v(x)$. (2) If $x \in \mathbf{x}_h$ is unassigned, then $UB(v) = \max_x P_v(x) \geq P_v(x^*)$. (3) If $y \in \mathbf{y}$ is marginalized out, then $UB(v) = \sum_y P_v(y) = 1$.

  From these cases, we observe that the upper bound computation potentially **overestimates** the contributions from leaf nodes associated with unassigned variables in $\mathbf{x}_h$, while the contributions from $\mathbf{x}_e$ and $\mathbf{y}$ remain unchanged. Once all variables are assigned, the leaf evaluations match those used in exact marginal inference.

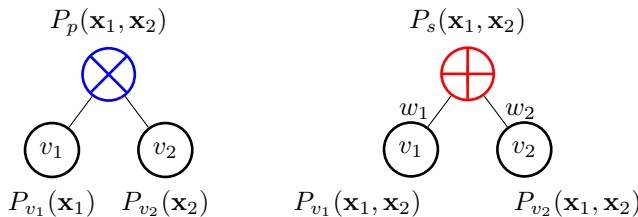

*Figure 8.* **(Left)** Example of a decomposable product node (colored blue). Denote the product node as $p$, and it has two children $v_1$ and $v_2$. Child nodes encode $P_{v_1}(\mathbf{x}_1)$ and $P_{v_2}(\mathbf{x}_2)$ respectively and the product node encodes $P_p(\mathbf{x}_1, \mathbf{x}_2) = P_{v_1}(\mathbf{x}_1)P_{v_2}(\mathbf{x}_2)$. Decomposability ensures $\mathbf{x}_1$ and $\mathbf{x}_2$ are disjoint. **(Right)** Example of a smooth sum node (colored red). Denote the sum node as $s$, and it has two children $v_1$ and $v_2$ with weights $w_1$ and $w_2$. Child nodes encode $P_{v_1}(\mathbf{x}_1, \mathbf{x}_2)$ and $P_{v_2}(\mathbf{x}_1, \mathbf{x}_2)$ respectively and the sum node encodes $P_s(\mathbf{x}_1, \mathbf{x}_2) = w_1 P_{v_1}(\mathbf{x}_1, \mathbf{x}_2) + w_2 P_{v_2}(\mathbf{x}_1, \mathbf{x}_2)$. Smoothness ensures all nodes encode probabilities over the same set of variables.

- **Induction step for a product node.** Suppose $v$ is a product node (see Figure 8 for an illustration). Without loss of generality, assume $v$ has two child nodes, $v_1$ and $v_2$, which represent $P_{v_1}(\mathbf{x}_e^{(1)}, \mathbf{x}_h^{(1)}, \mathbf{y}^{(1)})$ and $P_{v_2}(\mathbf{x}_e^{(2)}, \mathbf{x}_h^{(2)}, \mathbf{y}^{(2)})$, respectively.

  By the decomposability property, the scopes of $v_1$ and $v_2$ are disjoint, i.e., they do not share any variables. This allows the maximization over the product to be decomposed into the product of maximization over the children. More formally, we have

$$UB(v) = UB(v_1) \cdot UB(v_2) \geq \max_{\mathbf{x}_h^{(1)}} \sum_{\mathbf{y}^{(1)}} P_{v_1}(\mathbf{x}_e^{(1)}, \mathbf{x}_h^{(1)}, \mathbf{y}^{(1)}) \cdot \max_{\mathbf{x}_h^{(2)}} \sum_{\mathbf{y}^{(2)}} P_{v_2}(\mathbf{x}_e^{(2)}, \mathbf{x}_h^{(2)}, \mathbf{y}^{(2)})$$

$$= \max_{\mathbf{x}_h'} \sum_{\mathbf{y}^{(1)}} \sum_{\mathbf{y}^{(2)}} P_{v_1}(\mathbf{x}_e^{(1)}, \mathbf{x}_h^{(1)}, \mathbf{y}^{(1)}) P_{v_2}(\mathbf{x}_e^{(2)}, \mathbf{x}_h^{(2)}, \mathbf{y}^{(2)})$$

$$= \max_{\mathbf{x}_h'} \sum_{\mathbf{y}'} P_{v_1}(\mathbf{x}_e^{(1)}, \mathbf{x}_h^{(1)}, \mathbf{y}^{(1)}) P_{v_2}(\mathbf{x}_e^{(2)}, \mathbf{x}_h^{(2)}, \mathbf{y}^{(2)})$$

$$= \max_{\mathbf{x}_h'} \sum_{\mathbf{y}'} P_v(\mathbf{x}_e', \mathbf{x}_h', \mathbf{y}')$$

- **Induction step for a sum node.** Suppose $v$ is a sum node. Since the probabilistic circuit is smooth, all of its children must share the same scope of variables. Without loss of generality, assume $v$ has two child nodes, $v_1$ and $v_2$, which represent $P_{v_1}$ and $P_{v_2}$, with corresponding weights $w_1$ and $w_2$. Then we can derive

$$UB(v) = w_1 UB(v_1) + w_2 UB(v_2) \geq \max_{\mathbf{x}_h'} \left( \sum_{\mathbf{y}'} w_1 P_{v_1}(\mathbf{x}_e', \mathbf{x}_h', \mathbf{y}') \right) + \max_{\mathbf{x}_h'} \left( \sum_{\mathbf{y}'} w_2 P_{v_2}(\mathbf{x}_e', \mathbf{x}_h', \mathbf{y}') \right)$$

$$= \max_{\mathbf{x}_h'} \sum_{\mathbf{y}'} P_v(\mathbf{x}_e', \mathbf{x}_h', \mathbf{y}')$$

By recursive application of the update rules, we conclude that the upper bound satisfies $UB \geq \max_{\mathbf{x}_h} \sum_{\mathbf{y}} P(\mathbf{x}_e, \mathbf{x}_h, \mathbf{y})$ at the root node. A similar argument applies to the lower bound. Our proposed ULW algorithm follows this computation exactly, requiring only a single pass through the probabilistic circuit via bottom-up traversal. $\square$

## C. KOCO-SMC Implementation

Classical SAT solvers like MiniSAT (Eén & Sörensson, 2003) have achieved high performance in real-world applications. We implement our method based on MiniSAT version 2.2.0[2]. The decision, unit propagation, and backtracking steps are primarily derived from their implementation. We introduce the ULW algorithm, which enables conflict detection for probabilistic constraints in SMC problems. In this section, we present the implementation of the key components of KOCO-SMC. The pseudocode is shown in Algorithm 1.

---

[2]MiniSAT: https://github.com/niklasso/minisat

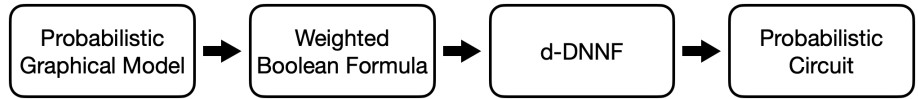

*Figure 9.* The process of constructing probabilistic circuits from probabilistic graphical models by ACE.

**Compilation** Koco-SMC requires that all probability distributions in the SMC problem be compiled into smooth and decomposable PCs. In our implementation, all distributions are represented as probabilistic graphical models, either in the form of Bayesian networks or Markov Random Fields.

The pipeline introduced in (Darwiche, 2002) (Fig. 9) compiles a distribution into a Boolean formula augmented with literal weights, which is then further compiled into a tractable algorithmic circuit. From this circuit, a tractable probabilistic circuit is derived. We use ACE[3] as the knowledge compilation tool.

**ULW Algorithm** After performing unit propagation on the Boolean constraints, we record the currently assigned variables at the current decision level. Next, we update the bounds for each affected probability distribution. These updated bounds are then compared to detect either a conflict or early satisfaction. If a probabilistic constraint is already satisfied, we mark it and skip further updates until a future backtracking step. If a probabilistic constraint becomes unsatisfiable, we raise a conflict using the same mechanism as for Boolean conflicts and record a corresponding conflict clause.

**Conflict Clause from the Probabilistic Constraint** The conflict clause derived from a probabilistic constraint is a Boolean clause that captures the cause of the conflict. We construct this clause by taking the negation of the current variable assignment. For example, if a constraint of the form $\sum_{x_3} P(x_1 = \texttt{True}, x_2 = \texttt{False}, x_3) > q$ triggers a conflict, we generate the Boolean clause $(\overline{x}_1 \vee x_2)$ as its negation. This ensures that any future assignment where $x_1 = \texttt{True}$ and $x_2 = \texttt{False}$ will violate the new clause, preventing the same unsatisfiable condition. Similar to CDCL-based SAT solvers, the learned clause is added to the Boolean formula so that the solver can avoid repeating the same conflict without needing to re-evaluate probabilistic inference.

## D. Experiment Setting

### D.1. Dataset Specification

All SMC problems in this study are in the form of $\phi(\mathbf{x}_\phi, \mathbf{x}_f) \wedge \left( \sum_{\mathbf{y}} f(\mathbf{x}_f, \mathbf{y}) > q \right)$ where $\phi(\mathbf{x}_\phi, \mathbf{x}_f)$ is a CNF Boolean formula, $f$ is a (unnormalized) probability distribution. $\mathbf{x}_\phi$ are decision variables that appear only in $\phi$, $\mathbf{x}_f$ are decision variables shared by $\phi$ and $f$, and $\mathbf{y}$ are latent variables that are marginalized.

**Boolean Formula** We pick random variables from $\phi$ and $f$ as shared variables uniformly at random. The number of shared variables between $\phi$ and $f$ (denoted as $\mathbf{x}_f$) is determined as the lesser of either half the number of random variables in $f$ or the total number of random variables in $\phi$; that is, the count of variables in $\mathbf{x}_f$ will not exceed either half the total number of variables in $f$ or the total number of variables in $\phi$.

All $\phi(\mathbf{x}_\phi, \mathbf{x}_f)$ represent 3-coloring problems for graphs, which involve finding an assignment of colors to the nodes of a graph such that no two adjacent nodes share the same color, using at most 3 colors in total. Each node in the graph corresponds to 3 random variables, say $x_1$, $x_2$, and $x_3$, where $x_1 = \texttt{True}$ if and only if the node is colored with the first color. We consider only grid graphs of size $k$ by $k$, resulting in $k \times k \times 3$ variables.

Those Boolean formulas are generated by CNFgen[4] using the command

```
cnfgen kcolor 3 grid k k -T shuffle
```

where the graph size $k$ is set to 5, 10, and 15. For each grid graph, we shuffle the variable names randomly and keep 3 of them.

---

[3]ACE: http://reasoning.cs.ucla.edu/ace
[4]CNFgen: https://massimolauria.net/cnfgen/

**Probability Distribution**   We use probabilistic graphical models from the UAI competition 2010-2022[5] including Markov random fields and Bayesian networks for the probabilistic constraints. Specifically, we pick the data for PR inference task, which includes 8 categories: Alchemy (2 models), CSP (3), DBN (6), Grids (8), ObjectDetection (79), Pedigree (3), Promedas (33), and Segmentation (6). The models with non-Boolean variables are removed, resulting in the remaining 50 models: *Alchemy* (1 model), *CSP* (3), *DBN* (6), *Grids* (2), *Promedas* (32), and *Segmentation* (6). All distributions are in the UAI file format. Since model counters d4, ADDMC, and SharpSAT-TD only accept weight CNF format in the model counting competition, we use bn2cnf[6] to convert data.

### D.2. Baselines

**Gibbs-SAA and BP-SAA**   are approximate SMC solvers based on Sample Average Approximation. The marginal probability in the form of $\sum_y P(x, y)$ is approximated using samples. More specifically, we use a sampler to generate a set of samples $\{(x, y^{(i)})\}$ according to a distribution proportional to $P(x, y)$. Then the marginal probability is estimated as the sample average $\frac{1}{N} \sum_{y^{(i)}} P(x, y^{(i)})$, multiplied by the number of possible configurations of $y$. For binary variables of length $n$, there are $2^n$ possible configurations. We use the Gibbs Sampler (Gibbs-SAA) and Belief Propagation (BP-SAA) implementations from (Ding & Xue, 2020) as the samplers. However, sampling is only an efficient probabilistic inference method, it still requires fixing $x$ in advance. Thus, we use Lingeling to enumerate solutions of $\phi(x)$.

Given a time limit of 1 hour, we set the number of samples to 10000 and the number of Gibbs burn-in steps to 40. For each SMC problem in the benchmark dataset, we run each approximate solver 5 times, and the problem is considered "solved" if at least one of those runs produces a correct result. The percentage of solved SMC problems is shown in Figure 4.

**XOR-SMC**   is an approximate solver from (Li et al., 2024). We set the parameter $T$ (which controls the probability of finding a satisfying solution—a higher $T$ yields better performance at the cost of longer runtime) to 3, and incrementally increase the number of XOR constraints from 0 until either a timeout or failure occurs. This process allows us to find the most probable satisfying solution. Similar to the SAA-based approaches, we run XOR-SMC 5 times.

**Lingeling-UAI10 and Lingeling-HBFS**   integrate the SAT solver Lingeling (Heule et al., 2019) with high-performance probabilistic inference solvers from the UAI Approximate Inference Challenge. The procedure begins by running Lingeling to produce a solution that satisfies the Boolean formula in the SMC problem. The resulting assignment is then passed to the inference solver to compute the corresponding marginal probability. If this probability exceeds the specified threshold, the solution is reported and the algorithm terminates. Otherwise, the SAT solver is invoked to generate a different solution, and the process is repeated until all solutions have been enumerated. To ensure a fair comparison, repetitive file I/O and solver initialization overheads have been minimized.

Lingeling-UAI10 uses the public inference solver LibDAI (Mooij, 2010), which participated in the UAI 2010 challenge and is available on GitHub[7]. Lingeling-HBFS uses the Toulbar2 solver (Cooper et al., 2010), which implements a hybrid best-first branch-and-bound algorithm (HBFS) for computing marginal probabilities. We use the public implementation of Toulbar2[8] for the probabilistic reasoning task, with default parameter settings.

**Lingeling-D4, Lingeling-ADDMC, and Lingeling-SSTD**   are integrations of the Lingeling SAT solver with the weighted model counting solver in the Model Counting Competition from 2020 to 2023. Lingeling-D4[9] uses d4 solver based on knowledge compilation. Lingeling-ADDMC uses the public implementation of the ADDMC solver [10]. Lingeling-SSTD uses SharpSAT-TD[11] as the model counter.

---

[5]UAI2022: https://uaicompetition.github.io/uci-2022/
[6]bn2cnf: https://www.cril.univ-artois.fr/KC/bn2cnf.html
[7]LibDAI: https://github.com/dbtsai/libDAI/
[8]Toulbar2: https://toulbar2.github.io/toulbar2/
[9]d4: https://github.com/crillab/d4
[10]ADDMC: https://github.com/vardigroup/ADDMC
[11]SharpSAT-TD: https://github.com/Laakeri/sharpsat-td

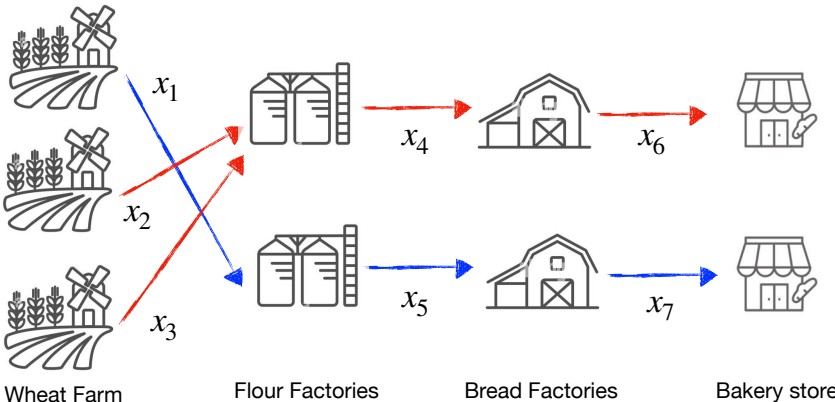

*Figure 10.* An example wheat to bread supply supply chain network. $x_i = \texttt{True}$ means the trade is selected from the supplier to the demander.

### D.3. Hyper-Parameter Settings

In all experiments, we use the public version of Lingeling implemented in PySAT[12] with their default parameter. The time limit for all approximate solvers (Gibbs-SAA, XOR-SMC) is set to 1 hour per SMC problem. The time limit for all exact solvers is 3 hours. All experiments are executed on two 64-core AMD Epyc 7662 Rome processors with 16 GB of memory.

### D.4. Application: Supply Chain Design

For the experiment on real-world supply chain network, we refer to a 4-layer supply chain network collected from real-world observations (Zokaee et al., 2017). An example is shown in Figure 10.

**Decision Variable.** Each edge between two nodes represents a trade, and the selection of trades can be encoded as a binary vector $\mathbf{x} \in \{\texttt{True}, \texttt{False}\}^M$, where $M$ is the number of edges. Here, $x_i = \texttt{True}$ indicates that the $i$-th edge (trade) is selected.

**Boolean Constraints.** Due to budget limitations, each node is assumed to receive raw materials from exactly 2 upstream suppliers and sell its products to exactly 2 downstream demanders.

**Probabilistic Constraints.** The supply chain design problem in (Zokaee et al., 2017) does not consider stochastic disasters; we address this by generating a Bayesian Network (BN) over all edges to model such random events. Let $P(x_1 = \texttt{True}, x_2 = \texttt{False}, \ldots)$ denote the joint probability that trade on edge 1 will not be affected by the disaster, trade on edge 2 will be affected, and so on. Suppose we choose to conduct trades only on edges 1 and 3. Then the marginal probability $P(x_1 = \texttt{True}, x_3 = \texttt{True}) = \sum_{x_2, x_4, \ldots} P(x_1 = \texttt{True}, x_2, x_3 = \texttt{True}, \ldots)$ denotes the likelihood that the selected trades are all successful.

Ensuring that the probability of the selected trades not being affected by disasters exceeds a certain threshold can be formulated as:

$$\sum_{\mathbf{x}_{\text{unselected}}} P(\mathbf{x}_{\text{selected}}, \mathbf{x}_{\text{unselected}}) > q$$

where we plan to execute trades on edges $\mathbf{x}_{\text{selected}}$. The marginal probability $\sum P$ corresponds to the likelihood that all selected trades are successfully conducted. To find the optimal plan, we incrementally increase the threshold $q$ from 0 to 1 in steps of $1 \times 10^{-3}$, continuing until the SMC problem becomes infeasible. The last feasible solution is referred to as the best plan.

**Construction of dataset.** This network consists of 4 layers of nodes representing suppliers, with each layer containing 9, 7, 9, and 19 nodes, respectively. Adjacent layers are fully connected, meaning each node can trade with any node in the neighboring layers (i.e., the nearest upstream suppliers and downstream demanders). We evaluate all exact SMC solvers on

---

[12]PySAT: https://pysathq.github.io/

three supply chain networks: a small network $[5, 5, 5, 5]$, a medium network $[7, 7, 7, 7]$, and a large network $[9, 7, 9, 19]$. The vector $[9, 7, 9, 19]$ represents the structure of the real-world network, with 9, 7, 9, and 19 suppliers in each layer, respectively. The other two networks are synthetic but of comparable scale. The results are shown in Figure 6.

For the specification of each generated disaster BN, each node can have at most 5 parents, and the number of BN edges is approximately half of the maximum possible. The generated BN is included in our code repository.

### D.5. Application: Package Delivery

For the case study of package delivery, our goal is to deliver packages to $N$ residential areas. We want this path to be a Hamiltonian Path that visits each vertex (residential area) exactly once without necessarily forming a cycle. The goal is to determine whether such a path exists in a given graph.

**Decision Variable.** Using an order-based formulation with variables $x_{i,j}$, where $x_{i,j}$ denotes that the $i$-th position in the path is occupied by residential area $j$, i.e., residential area $j$ is the $i$-th visited place.

$$x_{i,j} = \begin{cases} \text{True} & \text{if area } j \text{ is visited in the } i\text{-th position in the path,} \\ \text{False} & \text{otherwise.} \end{cases}$$

where the total number of variables is $N^2$ (for $N$ cities).

**Boolean Constraints.** $\phi$ is a CNF that checks if the variable assignment forms a Hamiltonian path. A Hamiltonian path in a graph is a path that visits each vertex exactly once.

**Probabilistic Constraints.** Additionally, we want the schedule to have a very high probability ($q$) of encountering light traffic.

$$P(x_{\text{path}}) = \sum_{x_{\text{env}}} P(x_{\text{path}}, x_{\text{env}}) \geq q$$

where $P$ denotes the probability of light traffic, $x_{\text{path}}$ represents the decision variables for the chosen path, and $x_{\text{env}}$ refers to latent environmental variables that affect traffic conditions, such as weather, road quality, and other external factors. The marginal probability $\sum_{x_{\text{env}}} P$ then represents the average probability of light traffic, marginalized over all possible environmental conditions.

**Construction of dataset.** The graph structures used in our experiments are based on cropped regions from Google Maps (Figure 11). We consider three sets of delivery locations: 8 Amazon Lockers, 10 UPS Stores, and 6 USPS Stores. The three maps we examine are: Amazon Lockers only (Amazon), Amazon Lockers plus UPS Stores (UPS), and UPS graph with the addition of 6 USPS Stores (USPS). These graphs consist of 8, 18, and 24 nodes, respectively.

The traffic condition probability is modeled by a Bayesian network (Figure 11) simplified from Los Angeles traffic data (West, 2020).

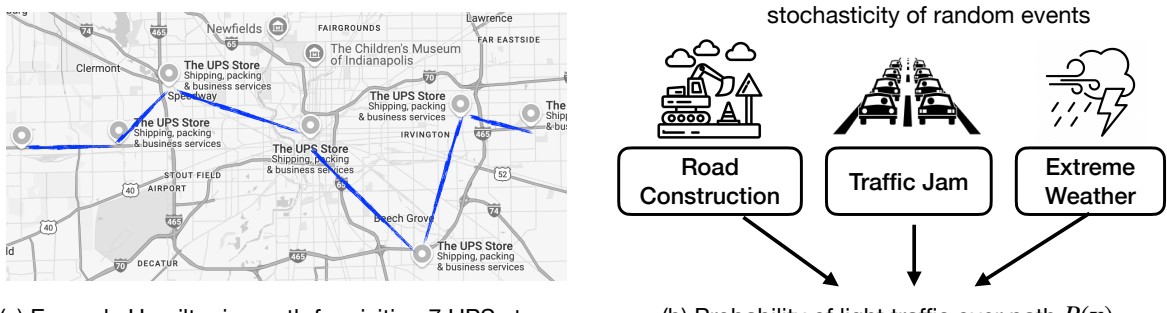

(a) Example Hamiltonian path for visiting 7 UPS stores.

(b) Probability of light traffic over path $P(\mathbf{x})$

*Figure 11.* Bayesian network for a single road (Hoong et al., 2012; West, 2020).

To find the best route, we gradually decrease the threshold of the probability of encountering heavy traffic from 1 to 0 in increments of $10^{-2}$, continuing until the threshold makes the SMC problem unsatisfiable. The running time for finding the best plan is shown in Figure 6(right).

# E. Additional Results

## E.1. Knowledge Compilation Time

The time for compiling graphical models to decomposable deterministic and smooth probabilistic circuits is shown in Figure 12.

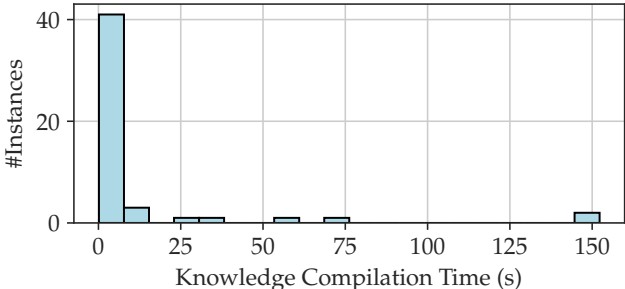

*Figure 12.* Histogram of the knowledge compilation time for all 50 probability distributions in the benchmark.

## E.2. Comparison with Exact Solvers

Figure 3(left) is an example shown in the main text. Additional results on other SMCs consisting of different Boolean formulas and probabilistic graphical models are shown below.

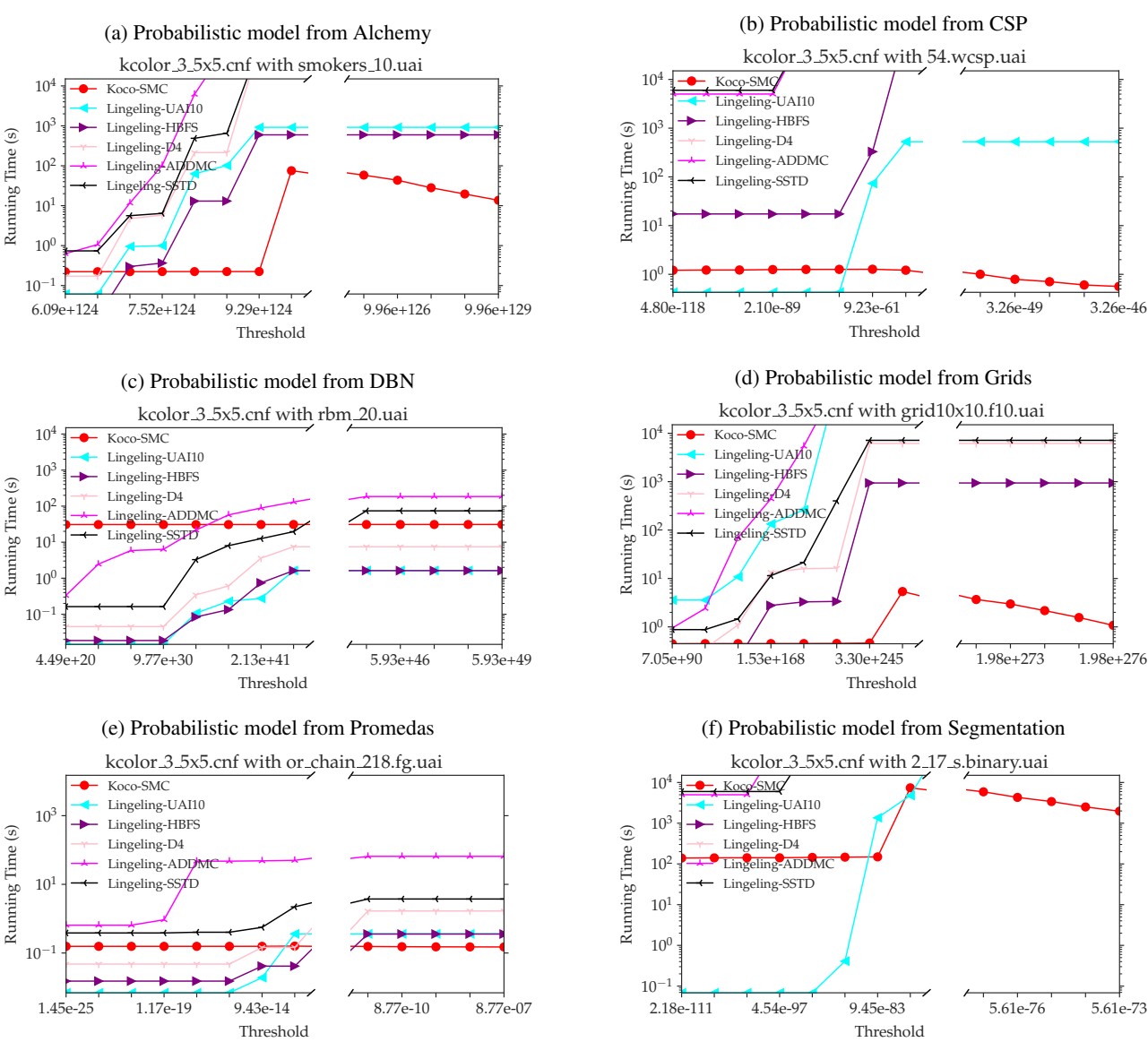

*Figure 13.* Results of SMC problems that consist of a fixed CNF file (*kcolor_3_5x5.cnf*) representing the 3 color problem on a $5 \times 5$ grid map and probabilistic graphical models from different categories.

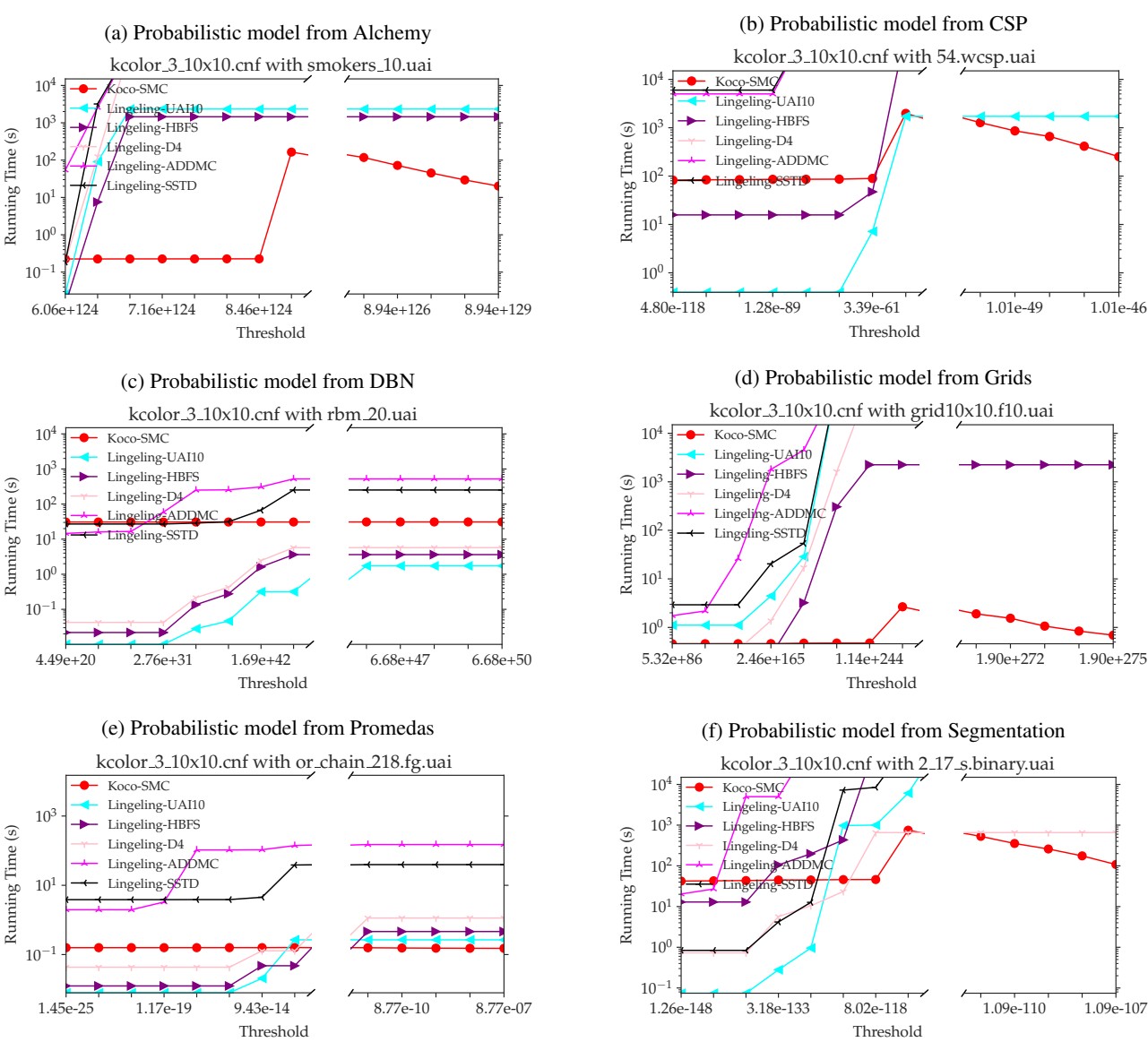

*Figure 14.* Results of SMC problems that consist of a fixed CNF file (*kcolor_3_10x10.cnf*) representing the 3 color problem on a $10 \times 10$ grid map and probabilistic graphical models from different categories.

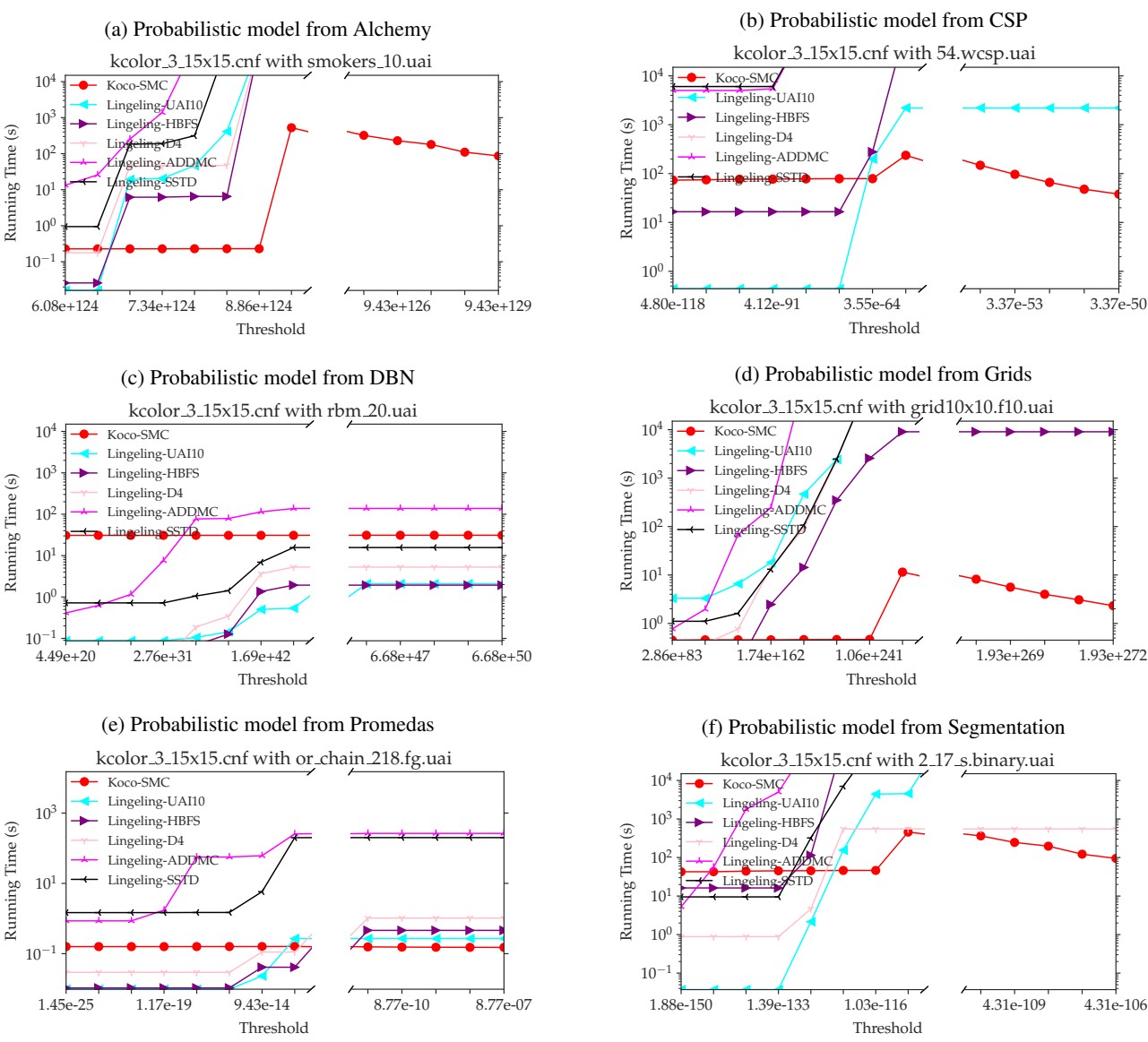

*Figure 15.* Results of SMC problems that consist of a fixed CNF file (*kcolor_3_15x15.cnf*) representing the 3 color problem on a $15 \times 15$ grid map and probabilistic graphical models from different categories.

