# OpenReview forum: "Solving Satisfiability Modulo Counting Exactly with Probabilistic Circuits"
_ICML.cc/2025/Conference — ICML 2025 poster_

### Official Review · Reviewer_L6SD · 2025-03-11

**Overall Recommendation:** 1

**Summary:**

This paper presents a new exact satisfiability-modulo-counting solver: `Koco-SMC`, and demonstrates its performance on benchmarks from a UAI benchmark set, compared to existing state-of-the-art exact and approximate SMC solvers.

The paper attempts to address a weakness in existing exact solvers: the need for them to alternate between a SAT solver and a probabilistic inference solver. It does so by incorporating an Upper Lower Watch (ULW) algorithm that keeps track of upper and lower bounds on the probabilistic inference parts of the problem, learning conflict clauses from those bounds as it goes.

The `Koco-SMC` pipeline relies on knowledge compilation to create a decision diagram that is converted into an arithmetic circuit, which is used for the inference. It incorporates branching heuristics, propagation and conflict-driven clause-learning techniques from the literature.

In the experimental evaluation `Koco-SMC` seems especially strong for problems where the probabilistic constraints are very hard to satisfy, or even result in unsatisfiable problems. `Koco-SMC` seems to refute unsatisfiable instance much faster than existing methods, due to its ability to detect conflicts instead of having to search a large search space.

**Claims And Evidence:**

I find that largely the claims about the experimental evaluation seem to be backed-up by the data. I do have some issues with the presentation of some of the data and the apparent conclusions that are drawn from it. I also feel that the paper may be missing some relevant literature, and therefore maybe also may be missing some SOTA in its evaluation. See comments below.

**Essential References Not Discussed:**

The reference for `CNFgen` is missing in the main text of the paper (and it only has a footnote in the appendix).

In my opinion, this paper should mention prior work on reasoning over constraints on the success probability of a literal-weighted CNF formula, and on the computation of upper and lower bound on the weighted model count of such a formula.

The following two works come from a line of work that presents algorithms for optimisation problems that require solving a constraint $C$ of the form $C := \text{Pr}(F(X,Y)) \bowtie \theta$, where $F(X,Y)$ represents a CNF on decision variables $X$ and random variables $Y$, where $0 < \theta \leq 1$ is a threshold on the success probability, and $\bowtie \in \{\leq, \geq\}$. They leverage existing MILP and CP technology to aid in branching, propagation and conflict learning, as well as creating a propagation algorithm specifically for probabilistic constraints on arithmetic circuits, allowing for the stochastic constraint to be combined with other constraints. Since they allow for multiple constraints of the above form to be added, and are similar in terms of approach (knowledge compilation, CNF, etc), I feel like they are close enough to the contents of this paper to merit at least a mention in the related work section, but probably a more detailed discussion or comparison:

A. L. D. Latour, B. Babaki, A. Dries, A. Kimmig, G. Van den Broeck, and S. Nijssen, ‘Combining stochastic constraint optimization and probabilistic programming — from knowledge compilation to constraint solving’, in _Proceedings of the 23rd international conference on principles and practice of constraint programming (CP 2017)_, in Lecture notes in computer science, vol. 10416. Springer, 2017, pp. 495–511.

A. L. D. Latour, B. Babaki, and S. Nijssen, ‘Stochastic Constraint Propagation for Mining Probabilistic Networks’, in _Proceedings of the Twenty-Eighth International Joint Conference on Artificial Intelligence_, Macao, China: International Joint Conferences on Artificial Intelligence Organization, Aug. 2019, pp. 1137–1145. doi: [10.24963/ijcai.2019/159](https://doi.org/10.24963/ijcai.2019/159).

The following work studies the use of knowledge compilation to derive lower and upper bounds on the success probability of a literal-weighted input Horn formula. Despite the title mentioning approximation guarantees, I believe the methods provided in this work also provide true weighted model counts. Given that the proposed work puts a lot of emphasis on the bounds computation, I feel that this literature should probably be mentioned:

A. Dubray, P. Schaus, and S. Nijssen, ‘Anytime Weighted Model Counting with Approximation Guarantees for Probabilistic Inference’, presented at the International Conference on Principles and Practice of Constraint Programming (CP), 2024. doi: [10.4230/LIPIcs.CP.2024.10](https://doi.org/10.4230/LIPIcs.CP.2024.10).

In the same spirit I think this work could merit a mention, since it concerns satisfiability modulo counting the number of solutions of a set of mixed-integer constraints, again keeping track of lower and upper bounds:

C. Ge and A. Biere, ‘Improved Bounds of Integer Solution Counts via Volume and Extending to Mixed-Integer Linear Constraints’, presented at the International Conference on Principles and Practice of Constraint Programming (CP), 2024. doi: [10.4230/LIPIcs.CP.2024.13](https://doi.org/10.4230/LIPIcs.CP.2024.13).

Finally, the following work also focuses on knowledge compilation for lower and upper bounds on the success probability of a literal-weighted input CNF, certifying the correctness of those bounds. This bit of literature is maybe a bit further from the contents of the proposed work, but I believe that it might still merit a mention:

C. Cheng, Y.-R. Luo, and J.-H. R. Jiang, ‘Knowledge Compilation for Incremental and Checkable Stochastic Boolean Satisfiability’, presented at the Thirty-Third International Joint Conference on Artificial Intelligence, Aug. 2024, pp. 1862–1872. doi: [10.24963/ijcai.2024/206](https://doi.org/10.24963/ijcai.2024/206).

I also find it strange that [Choi et al., 2022] is used as the reference for Smooth & Decomposability. Why not  cite

A. Darwiche, ‘On the Tractable Counting of Theory Models and its Application to Truth Maintenance and Belief Revision’, Journal of Applied Non-Classical Logics, vol. 11, no. 1–2, pp. 11–34, Jan. 2001, doi: 10.3166/jancl.11.11-34.

for smoothness and

A. Darwiche, ‘Decomposable negation normal form’, _J. ACM_, vol. 48, no. 4, pp. 608–647, Jul. 2001, doi: [10.1145/502090.502091](https://doi.org/10.1145/502090.502091).

A. Darwiche, ‘Compiling knowledge into decomposable negation normal form’, in _Proceedings of the sixteenth international joint conference on artificial intelligence, IJCAI 99, Stockholm, Sweden, july 31 - august 6, 1999. 2 volumes, 1450 pages_, T. Dean, Ed., Morgan Kaufmann, 1999, pp. 284–289. [Online]. Available: [http://ijcai.org/Proceedings/99-1/Papers/042.pdf](http://ijcai.org/Proceedings/99-1/Papers/042.pdf)

for decomposability? Or maybe the classic

A. Darwiche and P. Marquis, ‘A Knowledge Compilation Map’, _Journal of Artificial Intelligence Research_, vol. 17, pp. 229–264, Sep. 2002, doi: [10.1613/jair.989](https://doi.org/10.1613/jair.989)

?

If the papers must cite Choi et al. 2022, then please fix the currently incorrect representation of Guy Van den Broeck's name? It's rendered correctly in [Kisa et al., 2014].

I find in general that the bibliography is very sloppy. [Darwiche 1999] and [Darwiche 2002] have "Citeseer" in their bibliography entries. The reference for [Shet et al. 2023] is a preprint, even though that work also seems to have a published version. There is some weirdness going on in the pdf where there are a lot of instances of empty space hyperlinking to the bibliography entry for [Li et al., 2024], which itself does not list page numbers. A lot of titles are rendered incorrectly, with letters that should be in uppercase being rendered as lower-case, instead. Furthermore, some items use abbreviated venue titles, while others write them out fully.

**Experimental Designs Or Analyses:**

I find some of the claims insufficiently supported, in part because I find the way in which they are presented somewhat misleading.

An example is Figure 3. Its caption claims that "Koco-SMC solves 80% of SMC problems in 20 ... ". However: as I understand the text, the figure shows the result for one problem ("specific combination (3-color-5x5.cnf with smokers_10,uai)", the meaning of which is also unclear to me) for different thresholds $q$. This implies to me that the problem structure (and thus the compiled probabilistic circuit) remains the same. As I understand it, the experiments show running time including the compilation time. Given that compilation time can certainly vary quite a lot from instance to instance (and are independent of threshold $q$), I find that presenting those times for just one problem instance (for varying $q$) a bit reductive. Furthermore, the claim that "our method requires significantly less time across most instances" does not seem to be backed up by any statistical tests that would demonstrate statistical significance, and it is not clear to me what this sentence even means (less time than what? what does "across most instances" mean?).

I do think that reading Appendix C in detail would inspire trust in the quality of the empirical evaluation. However, my above comments on how I find the claims related to Figure 3 somewhat misleading, stands.

**Methods And Evaluation Criteria:**

Based on the main text, I do not understand how the benchmark instances were selected for the experiment. I cannot find a URL to download the instances, or a reference to which instances were used, exactly. I have no idea what "50 models over binary variables are kept." means in this context?

As I understand it, the benchmarks are created by somehow Frankensteining probabilistic constraints and SAT formulae together, but it is not clear to me how.

I do like that the description of the used solvers is finally made quite explicit in Section 5.1.

**Other Comments Or Suggestions:**

lines 47-48: I'm a bit surprised to find only references to workshops here, not to actual literature, except for Sheth et al 2023, which is a reference to a preprint even though a published version of that same work is also available.
- Figure 2: poor formatting. Subfigures are not actually formatted as such, which results in a mismatch of fonts and an inability to search the figure for text.
- line 80: I would really like to see references here so I know which "Current exact SMC solvers" the paper is referring to.
- line 90: why is there a hyperlink to the [Li et al., 2024] entry in the bibliography at the start of this paragraph?
- lines 102-103: what is meant with "the largest dataset based on the UAI Competition benchmark"? Could this be made more specific? As I understand it, each edition of UAI has a different benchmark set? It would be helpful if the paper was more transparent about which benchmarks were used exactly, and if there was a reference here.
- lines 105-106: It would help my understanding if the paper explained the notation of $\{f_i\}^{K}_{i=1}$. What is $K$? What is $i$? Why does $i=1$ need to be there?
- Lines 108-110: Is it essential that $|\mathbf{x}| = |\mathbf{y}_i| = |\mathbf{z}_i|$? Why or why not? Are there any constraints on the value of $L$?
- Line 143: "Formally, PC" missing indefinite article.
- Line 157: "The root node $r$ in the graph has no parent node." This confuses me. Seems obvious, right? Why does it need specifying?
- The use of the word "model" is somewhat overloaded in the paper. It seems that in line line 137 it means "a satisfying assignment to a Boolean satisfiability formula", in line 158 it means "a representation of a probability distribution", and in line 315 it means "problem instance"? Do I understand this correctly? Could the writing be improved by being more explicit about this overloading?
- Line 151: why is there no part (d)?
- Line 155 "An existing exact SMC" solver: Which one? Can I have a reference? Or is the intended meaning "any existing exact SMC solver that we are aware of"?
- Line 185: "SMC solver" missing "s".
- Line 350: I find "facing SMC problems" a very awkward description of what is going on.
- Line 352: Should this and the following $Q$s be $q$s, instead?
- Line 355-356: I have no idea what "specific combination (3-color-5x5.cnf with smokers_10,uai)" means.
- I will die on this hill: an x-axis is only an x-axis if it is labelled with "x". Otherwise, it is likely a horizontal axis, or a time axis, or in this case maybe s $q$-axis, or whatever is appropriate. Similar hill for y-axis.
- There are spaces missing here and there after Figure numbers.

**Other Strengths And Weaknesses:**

I really like the detailed case studies that are described in the paper. I also want to compliment the authors on their efforts to make the line graphs b+w printer-friendly. The paper contains few typos, and I find that it reads well in general.

**Questions For Authors:**

Q1. Can the authors please reflect on how their work relates to the work presented in [Latour et al., 2017], [Latour et al., 2019], and [Dubray et al., 2024] (see above)? As I currently understand the paper, I feel like these are relevant enough to be mentioned, or maybe even compared against. For now, this is a reason for me to recommend rejection.

Q2. Can the authors please elaborate on their claim that "Our method typically requires significantly less time across most instances."? As argued above, I find this claim, especially in the context of Fig 3 somewhat misleading. As I explained above, I do not know what it means, which for now also makes me less eager to recommend accepting the work.

**Relation To Broader Scientific Literature:**

As I understand it, the main contribution improves on the reported state of the art by learning conflicts directly from the probabilistic constraints, instead of alternating between a SAT solver and a solver for probabilistic inference. In my opinion, the paper misses some of the relevant literature, see remarks below.

**Theoretical Claims:**

There is a lemma, with a sketch of the proof. I'm not sure that a lemma is needed here. I find that it is phrased somewhat awkwardly ("equality *can* be achieved"? so sometimes it isn't?), and I feel like it does not add much compared to the existing (cited) literature on computing probabilities with arithmetic circuits. Hence, in my opinion this lemma does not add much. I would rather want to see the space it takes up now used to discuss more of the existing literature, or to clarify certain parts of the implementation or empirical evaluation (see also my notes elsewhere in this review on those topics).

---

> ### Author Rebuttal · Authors · 2025-03-30
>
> Thank you very much for the detailed review. We will address your concerns and questions in the following reply.
>
> ---
> ### **Q1: Related works**
>
> We deeply appreciate your effort in pointing out the lines of work we missed. We have carefully reviewed the related literature and summarized the main changes below:
>
> 1. Relavance to upper and lower bound computation: [Link](https://anonymous.4open.science/r/anonym_koco_smc-61FD/plot/related-bounds.png)
>
> 2. Relavance to Stochastic Constraint Optimization Problems: [Link](https://anonymous.4open.science/r/anonym_koco_smc-61FD/plot/related-optimization.png)
>
> 3. Modified citations for PCs: [Link](https://anonymous.4open.science/r/anonym_koco_smc-61FD/plot/related-pc.png)
>
> ---
>
> ### **Q2: SMC problem definition**
>
> 1. What is {$f_i$}$_{i=1}^K$?
>
> It means a set of functions {$f_1,\ldots, f_K$}. K is the number of total probabilistic constraints.
>
> 2. Is it essential that $|x|=|y_i|=|z_i|$?
>
> No, they are not, because x represents the decision variable, and y, z are marginalized-out latent variables. Their lengths can be arbitrary. We revised the notation in our problem formulation as follows and hope this answers your question.
>
> [Link to the revised version](https://anonymous.4open.science/r/anonym_koco_smc-61FD/plot/problem-formulation.png)
>
> ---
>
> ### **Q3: Lemma 1 seems redundant**
>
> Our Lemma 1 holds, or equivalently, Equation 3 holds, when the PC satisfies the smoothness and decomposability properties. Otherwise, the computed upper and lower bounds (UB and LB) through our procedure may not be valid; that is, Equation 3 may be violated for some value assignments to the remaining variables. Thus, it is necessary to place Lemma 1 in the main text.
>
> ---
>
> ### **Q4: Experiments**
>
> 1. Dataset details
>
> Each SMC instance consists of a CNF file for the Boolean SAT constraint, a UAI file for the probabilistic constraint, and a threshold value.
>
> The original CNF files (9 files generated by CNFGen) and UAI files (50 files selected from UAI competitions held from 2010 to 2022, see details in Appendix C.4) are in the anonymous repository:
>
> `https://anonymous.4open.science/r/anonym_koco_smc-61FD/data/`
>
> The threshold values are available in the file: `anonymous.4open.science/r/anonym_koco_smc-61FD/data/benchmark_ins/data/inst_[cnf name]_[uai name]/thresholds.txt`
>
> For the compiled probabilistic circuit files,  most of them are excessively large (>1 GB), so we didn’t include them in the repository for now.
>
> 2. Detailed settings for Figure 3
>
> In Figure 3, we conduct an experiment on a single instance, where the CNF uses the 3-color5x5.cnf file (representing a 3-coloring problem on a 5×5 grid map), and the probabilistic distribution uses the smokers-10.uai file (from the UAI 2012 Competition), with various threshold $q$. The running time of our approach includes the knowledge compilation time for a fair comparison. We conduct extended experiments in Figures 14, 15, and 16.
>
> 3. Regarding 'our method requires significantly less time across most instances'
>
> According to the results in Figures 3, 14, 15, and 16, we find our method is more efficient than baselines in most cases. In some cases, like Figure 14(c), our method shows inferior running time performance due to huge compilation overhead, which agrees with your statement.
>
> ---
>
> ### **Q5: Clarification on “Current exact SMC solvers (line 80)” and "An existing exact SMC solver (line 155)"**
>
> We realize that these phrases are misleading as they suggest the existence of an actual tool. To clarify, we have revised the text accordingly.
>
> ```
> Since there is no general exact SMC solver, solving SMC problems exactly requires combining tools from SAT solving and probabilistic inference.
> ```
>
> ---
>
> ### **Q6: Other issues**
>
> 1. On line 90, we didn’t see the hyperlink in the PDF. We would appreciate it if you could provide more context.
>
> 2. Bibliography issue.
>    We've carefully revised the bibliography. [Link to new Bibliography](https://anonymous.4open.science/r/anonym_koco_smc-61FD/plot/references.png)
>
> 3. Language and writing issues: We have cleared up the improper use of language errors according to your suggestions.
>
> 4. Line 157: "The root node  in the graph has no parent node."
>    It is changed to:
>
>    “The root node is the final output of the probabilistic circuit. It represents the overall probability distribution over the variables.”
>
> 5. The use of the word "model" is somewhat overloaded in lines 137, 158, and 315.
>    We have cleaned up the usage of the word “model” in these lines for clarity.
>
> 6. The use of the x-axis and y-axis for Figure 3 caption.
>    We have cleaned up the related text. It should be:
>
>    [Link to the revised Figure 3](https://anonymous.4open.science/r/anonym_koco_smc-61FD/plot/figure3.png)
>
> ---
>
> Thank you again for your detailed review—your comments have been incredibly helpful in improving the paper. We hope this addresses your concerns, and we’d be happy to discuss further if you have any additional feedback.

---

> > ### Comment · Reviewer_L6SD · 2025-04-07
> >
> > Dear authors,
> >
> > Thank you for your detailed rebuttal. Here are some follow-up questions regarding your answers:
> >
> > Q2, related works point 2: Can the authors elaborate on what they mean by "Our KOCO-SMC is more specialized for this kind of problem"? Specialised how? What exactly are the authors referring to with "this kind of problem"?
> >
> > Q6, point 1: the point is that the hyperlink is invisible until you "mouse-over" it. It happens when I mouse-over the left side of the "W" that starts line 90.
> >
> > Q6, point 2: bibliography still messy. Still faulty rendering of Guy Van den Broeck's name in IJCAI 2020 workshop reference. Should be "Van den Broeck, Guy". Didn't check the rest.
> >
> > Kind regards

---

> > > ### Author Response · Authors · 2025-04-07
> > >
> > > Dear reviewer,
> > >
> > > We deeply appreciate your comments.
> > >
> > > ---
> > >
> > > ### **Q2 related works**
> > >
> > > **Connection between our method and existing works:** Stochastic Constraint Optimization Problems (SCOPs) have an intrinsic connection with SMC problems. This connection enables the transformation of SMC problems into SCOPs, which can then be reformulated as Mixed-Integer Linear Programs (MILP) or Constraint Programming (CP) problems. Such reformulations allow us to take advantage of powerful, off-the-shelf solvers that utilize techniques like branch-and-bound to solve these problems efficiently.
> > >
> > > **Our novelty:** KOCO-SMC is an exact solver tailored for general SMC problems, combining SAT solvers with probabilistic inference techniques. Our experiments specifically demonstrate the effectiveness of the ULW algorithm compared with vanilla baselines.
> > >
> > > In future work, we plan to compare our approach with MILP and CP solvers to identify the strengths and weaknesses in solving SMC problems.
> > >
> > > ---
> > >
> > > ### **Q6 point 1**
> > >
> > > Thank you for the detailed explanation. We have identified the issue—it was caused by a template problem on line 74, where the reference (Li et al., 2024) was split across two pages. We've fixed it accordingly.
> > >
> > > ---
> > >
> > > ### **Q6 point 2**
> > >
> > > Thank you for confirming the name "Van den Broeck, G." We identified that the issue originated from the DBLP biography with the auto-formatting style. We have corrected the name using the proper format and have reviewed all other information as well.
> > >
> > > ---
> > >
> > > We sincerely thank you for your response. We hope this addresses all of your concerns. Please don’t hesitate to let us know if there's anything further you'd like to discuss—we would greatly appreciate the opportunity to continue the conversation.
> > >
> > > Best regards

---

### Official Review · Reviewer_HcdQ · 2025-03-14

**Overall Recommendation:** 4

**Summary:**

This paper investigates the Satisfiability Model Counting problem, an extension of SAT that incorporates constraints involving probabilistic inference. The authors propose, KOCO-SMC, an efficient exact SMC solver that leverages probabilistic circuits through knowledge compilation to accelerate repeated probability estimation. To enhance efficiency, they introduce the ULW algorithm, which monitors lower and upper bounds of probabilistic inference for early conflict detection. Empirically, KOCO-SMC significantly outperformed existing exact and approximate SMC solvers on real-world problems.

## update after rebuttal
I thank the authors for their response and maintain my acceptance recommendation.

**Claims And Evidence:**

The properties of the ULW algorithm are supported by theoretical analysis, while empirical evaluations confirm the superior runtime performance of KOCO-SMC compared to existing methods.

**Essential References Not Discussed:**

Citations are missing in the Probabilistic Inference and Model Counting section of the related works.

**Experimental Designs Or Analyses:**

I reviewed the experimental designs for the UAI dataset, the two real-world problems, and the ablation study for ULW. The designs and analyses appear sound to me.

**Methods And Evaluation Criteria:**

The proposed methods behind KOCO-SMC, along with its empirical evaluation, are well-suited for SMC problems.

**Other Comments Or Suggestions:**

Please add citation to CaDiCaL.

At line 200, "into" is missing between “probabilistic constraints” and “probabilistic circuits”.

**Other Strengths And Weaknesses:**

N.A.

**Questions For Authors:**

Beyond conflict detection, is KOCO-SMC able to derive propagations from probabilistic inference constraints?

**Relation To Broader Scientific Literature:**

The SMC problem can encode both symbolic and probabilistic constraints and has found applications in real-world scenarios. This work introduces an efficient exact SMC solver, advancing the practical applicability of SMC problems. Previous works have primarily focused on approximate methods or relied on a straightforward combination of a SAT solver and a probabilistic inference solver, often resulting in poor efficiency.

**Theoretical Claims:**

I have verified the claim in Lemma 3.1, and it is correct.

---

> ### Author Rebuttal · Authors · 2025-03-30
>
> Thank you very much for your review and suggestions.
>
> ---
> ### **Q1: Beyond conflict detection, is KOCO-SMC able to derive propagations from probabilistic inference constraints?**
>
> This is a very good question. In our context, propagation specifically refers to unit propagation, which derives new variable assignments based on current assignments. Currently, we only perform unit propagation on the Boolean constraints and not on the probabilistic constraints.
>
> The reason is as follows: consider the constraint $\sum_y f(x_1, x_2, x_3, y) > q$. Suppose $x_1$ and $x_2$ have already been assigned, and $x_3$ remains unassigned. Since $f(\cdot)$ is represented as a complex probabilistic circuit (PC), we cannot directly derive the assignment for $x_3$. Instead, we need to evaluate both cases—assigning $x_3$ as True and False—because this assessment can yield three possible outcomes: (1) Both assignments (True and False) satisfy the constraint. (2) Only one assignment satisfies the constraint. (3) Neither assignment satisfies the constraint.
>
> ---
>
> ### **Q2: Add Citations**
>
> We have added a citation to CaDiCaL and included additional references in the "Probabilistic Inference and Model Counting" section:
>
> [1] Chavira, Mark, and Adnan Darwiche. "On probabilistic inference by weighted model counting." *Artificial Intelligence* 172.6-7 (2008): 772-799.
>
> [2] Cheng, Qiang, et al. "Approximating the sum operation for marginal-MAP inference." *Proceedings of the AAAI Conference on Artificial Intelligence*. Vol. 26, No. 1, 2012.
>
> [3] Gomes, Carla P., Ashish Sabharwal, and Bart Selman. "Model counting: A new strategy for obtaining good bounds." *AAAI*. Vol. 6, 2006.
>
> [4] Achlioptas, Dimitris, and Panos Theodoropoulos. "Probabilistic model counting with short XORs." *International Conference on Theory and Applications of Satisfiability Testing*. Springer International Publishing, Cham, 2017.
>
> ---
>
> ### **Q3: Writing typos**
>
> We have added the missing word "into" at line 200.
>
> ---
>
> ### **Q4: Ethical Review Flag**
>
> We noticed that you flagged this paper for an ethics review. We wanted to confirm whether this was done by accident. We would greatly appreciate further details to properly address any concerns.
>
> ---
>
> We greatly appreciate your insightful comments. If you have any additional suggestions or questions, please feel free to let us know—we are happy to address them.

---

> > ### Comment · Reviewer_HcdQ · 2025-04-09
> >
> > The ethics flag was applied by mistake. I have removed it.

---

### Official Review · Reviewer_1wbr · 2025-03-17

**Overall Recommendation:** 3

**Summary:**

This paper aims to provide an efficient solution to the satisfiability modulo counting problem. It proposes to use probabilistic circuits to encode the propositional formula which is further combined with a conflict-driven clause learning framework to compute bounds for the marginal distributions. Empirical evaluations on the resulting algorithm are further presented.

**Claims And Evidence:**

At the end of Section 3.1, the paper claims that the current solvers suffer from the fact that they need to go back and forth invocation between SAT solver and the probabilistic inference process. Still, in the proposed algorithm solver Koco-SMC, it uses the CDCL framework and there's still the back-and-forth since when an unsat assignment comes up, it augments the PC with the unsat clauses just like what the baseline does. Further, since the PC is required to be both smooth and decomposable in this framework, the process of adding learned clauses might introduce non-negligible computational cost. It is unclear to me why the proposed framework is more efficient.

**Essential References Not Discussed:**

This work is closely related to the field of weighted model counting where some of the WMC algorithms are also CDCL [1] based but no reference is presented.

[1] Möhle, Sibylle, and Armin Biere. "Combining Conflict-Driven Clause Learning and Chronological Backtracking for Propositional Model Counting." GCAI. 2019.

**Experimental Designs Or Analyses:**

The experimental results seem sound and the proposed algorithm outperforms the baseline methods.

**Methods And Evaluation Criteria:**

I don't fully understand the problem formulation of SMC: 1) What are the differences between variables x, y and z in Equation (1) and (2)? 2) It is unclear what the given probabilities are but only the constraints on the marginal distributions are defined as in Equation (1) and (2). Are some probability tables also defined in the formulation of SMC as to the conditional probability tables in Bayesian networks? (3) I don't understand the initialization as described at the end of Section 3 due to the previous two confusions. (4) How is the 0.1 at Line 165 obtained?

**Other Comments Or Suggestions:**

N/A

**Other Strengths And Weaknesses:**

N/A

**Questions For Authors:**

See Methods And Evaluation Criteria.

- What is the time complexity required to add the learned clauses to PCs? Would it blow up the size of PCs?
- It seems only when the upper bound is lower than q or the lower bound is greater than q, there would be definite outcomes for satisfiability. What if it's neither case? Would it result in approximate solutions instead of being an exact solver?

**Relation To Broader Scientific Literature:**

It contributes to the field of statistical machine learning.

**Theoretical Claims:**

Not applicable.

---

> ### Author Rebuttal · Authors · 2025-03-30
>
> Thank you for your careful review and for raising valuable questions.
>
> ---
>
> ### **Q1: Why is Koco-SMC More Efficient?**
>
> In general, KOCO-SMC saves time by detecting conflicts early using partial variable assignments, whereas baseline solvers require full variable assignments.
>
> - **Baseline Solvers:** These solvers require multiple back-and-forth interactions between SAT-solving and probabilistic inference. Typically, they first assign **all decision variables** using the SAT solver before checking probabilistic constraints. When a conflict occurs, baseline solvers can only learn to avoid repeating that specific complete assignment.
> - **Koco-SMC:** In contrast, Koco-SMC performs back-and-forth checks after assigning **just a single variable**. It immediately updates the upper and lower probability bounds using the efficient ULW algorithm to detect conflicts. This approach allows conflicts to be discovered much earlier—after only partial assignments—thus avoiding unnecessary evaluations of remaining variables.
>
> **Extreme Example:** When probabilistic constraints in an SMC problem are inherently unsatisfiable, Koco-SMC can confirm unsatisfiability by assigning just a single decision variable. Baseline solvers, however, must enumerate and test all potential assignments generated by the SAT solver.
>
> ---
>
> ### **Q2: Clarification of the Problem Formulation**
>
> **(1) What are the differences between variables x, y, and z in Equations (1) and (2)?**
>
> The variables **x** denote the decision variables, while **y** and **z** are latent variables that are marginalized out.
>
> **(2) It is unclear what the given probabilities are in Equations (1) and (2).**
>
> The given probability distributions can be any weighted functions mapping Boolean variables to real numbers—for example, probabilistic circuits, Markov random fields, or Bayesian networks. The SMC problem formulation imposes no specific constraints on these probability distributions.
>
> For KOCO-SMC, we specifically compiles probability distributions into probabilistic circuits because PCs have structural properties that significantly benefit our ULW algorithm.
>
> **(3) I don't understand the initialization as described at the end of Section 3 due to the previous two confusions.**
>
> The initialization mentioned in line 226 refers to the computation of the initial upper and lower probability bounds within the PC.
>
> **(4) How is the value 0.1 at line 165 obtained?**
>
> This value is provided as an illustrative example demonstrating why our method is more efficient. Detailed steps for this calculation are included in Appendix Figure 7.
>
> Specifically, line 165 illustrates the calculation using the example distribution represented by the probabilistic circuit in Figure 2(c). By setting **x₁** and **x₂** to **True**, we can compute the marginal probability over **x₃** and **x₄** as **0.1**.
>
> ---
>
> ### **Q3: Answers to Specific Questions**
>
> - **What is the time complexity required to add the learned clauses to PCs?**
>
>     The time complexity for adding learned clauses is constant. Our approach adds learned clauses only to the Boolean formula to prevent repeated conflicts. The structure of the PCs remains unchanged.
>
> - **What if neither bound condition is met? Would it result in approximate solutions rather than an exact solver?**
>
>     No, it would not lead to approximate solutions. Instead, it implies that additional variables must be assigned to narrow down the upper and lower bounds.
>
>     Consider a constraint such as $\sum_y f(x,y) > q$. If the lower bound of $\sum_y f(x,y)$ is greater than $q$ (or the upper bound is less than $q$), we can conclude UNSAT (or SAT). Otherwise, we continue assigning variables. Once all variables are assigned, $\sum_y f(x,y)$ will yield an exact numeric value without ambiguity. Thus, Koco-SMC remains an exact solver.
>
> ---
>
> ### **Q4: Other Issue**
>
> Thank you for suggesting the WMC algorithms based on CDCL. We have now incorporated these references and other relevant related work into the paper.
>
> ---
>
> Thank you again for your constructive comments. If there are any additional points you'd like us to clarify or discuss, please feel free to let us know.

---

### Official Review · Reviewer_Vn9u · 2025-03-17

**Overall Recommendation:** 4

**Summary:**

Satisfiability modulo counting (SMC) is a generalisation of SAT that consists of a propositional formula phi(x,b) and a collection of statements of the forms

1) the marginalisation of a discrete probability function f(x,y) (marginalising the variables in y) is at least some constant q,

2) the marginalisation of a discrete probability function f(x,y) (marginalising the variables in y) is at least the marginalisation of a discrete probability function g(x,z) (marginalising the variables in z).

The truth value of statements of the form 1) and 2) determine the values of the boolean variables in b.

That is, SMC instance asks whether values for x can be give such that phi(x,b), where the variables in b are interpreted as the truth values of statements of the form 1) or 2).

The authors introduce an exact solver for SMC, which they call Koco-SMC. The authors claim that previous approaches to integrate SAT solver to this task boil down to having a SAT solver to enumerate all possible solutions to phi(x,b), and then using a probabilistic inference solver to check whether the constraints 1) and 2) hold, as dictated by the selection of the variables in b. The approach that the authors take in Koco-SMC is to use standard tools for CSPs (e.g., propagation, conflicts clause learning, backtracking) to pick partial assignments to variables in phi(x,b) and then to check whether those partial assignments for variables yield a contradiction for the probability statements. The authors construct probabilistic circuits for computing the value of the functions mentioned in 1) and 2). The use of such functions allows the authors to efficiently compute upper and lower bounds for the values of the functions under partial assignments. These lower and upper bounds can be then used to infer contradiction regarding the probability statements.

Finally the authors experimentally compare their solver to other approximate and exact solver for solving SMCs. Their findings show that their approach is more efficient than others in the literature.

## update after rebuttal
I maintain my evaluation.

**Claims And Evidence:**

The paper is well written and presented. The claims given are supported.

**Essential References Not Discussed:**

I am not aware of related works that should be cited.

**Experimental Designs Or Analyses:**

I did not check details related to the experiments.

**Methods And Evaluation Criteria:**

The authors give a comprehensive comparison of their approach to others in the literature. The methods and evaluation criteria seems suitable for the task.

**Other Comments Or Suggestions:**

None.

**Other Strengths And Weaknesses:**

I think the main abstract contribution of the paper is to use probabilistic circuits as an efficient method to compute upper and lower bounds for the probabilistic functions, which can then be used to yield contradictions. This approach also requires the use of partial assignments, but there the use of CSP techniques are quite standard. Otherwise, from the theoretical point of view, the paper is not that deep. However, if this approach is indeed novel, in this context, it a valuable contribution. Also the experimental results show that their approach work better in practice than the previous approaches.

**Questions For Authors:**

None.

**Relation To Broader Scientific Literature:**

The authors do a good job in setting the scene of the paper and in positioning their results with respect to literature.

**Theoretical Claims:**

The description of their approach in the main part of the paper seems plausible and correct. I did not check the correctness of all the claims from the appendix.

---

> ### Author Rebuttal · Authors · 2025-03-30
>
> Thank you very much for your review and your positive assessment of our paper. We appreciate your concise summary of our contributions and your acknowledgment of the practical value demonstrated by our experimental results.
>
> ---
>
> ### **Q1. Main Contribution**
>
> We propose an integrated **exact SMC solver**, addressing the existing gap in this research community, which previously only had approximate solvers available.
> Our method achieves higher computational efficiency compared to vanilla exact solvers by utilizing our proposed ULW algorithm.
> Vanilla exact solvers directly combine SAT solving and probabilistic inference processes, resulting in slow performance due to frequent back-and-forth invocations between these two solvers.
>
> ---
>
> ### **Q2. Novelty of Our ULW Algorithm**
>
> Our ULW algorithm is inspired by CSP, SAT, and optimization literatures. We have revised the related work section to further clarify the novelty of our contributions:
>
> 1. **Relevance between ULW and Upper and Lower Bound Computation**:
>
> ```
> Our KOCO-SMC efficiently tracks upper and lower bounds for probabilistic constraints with partial variable assignments. In literature, (Dubray et al., 2024; Ge & Biere, 2024) compute approximate bounds for probabilistic constraints based on the DPLL algorithm. (Choi et al., 2022; Ping et al., 2015; Marinescu et al., 2014) provide exact bounds by solving marginal MAP problems, yet they are more time consuming than our method.
> ```
>
> 2. **Relevance between SMC and Stochastic Constraint Optimization Problems**:
>
> ```
> Stochastic Constraint Optimization Problems (Latour et al.,2019; 2017) can be formulated as MILP or CP problems using stochastic constraints to solve SMC problems. Our KOCO-SMC is more specialized for this kind of problem.
> ```
>
> The related works above bridge the connection to the existing theoretical foundations.
>
> ---
>
> Thank you once again for your valuable feedback. Please let us know if you have any further concerns, and we would be happy to discuss them.

---

### Decision · Program_Chairs · 2025-05-01

**Decision:**

Accept (poster)

**Comment:**

This paper addresses the Satisfiability Modulo Counting (SMC) problem, which extends SAT by incorporating probabilistic constraints. The authors propose KOCO-SMC, an exact solver that uses probabilistic circuits to efficiently compute bounds for probability statements within a conflict-driven clause learning framework. Experimental results demonstrate that KOCO-SMC outperforms existing exact and approximate SMC solvers across various benchmarks.

The authors were largely positive about the paper. The sole negative reviewer did not further engage in the discussion which is taken as an indication of agreement to the other reviews.

Hence I recommend accepting the paper.